# GEPC: Group-Equivariant Posterior Consistency for Out-of-Distribution Detection in Diffusion Models

**Yadang Alexis Rouzoumka** [1][2]  **Jean Pinsolle** [2]  **Eugénie Terreaux** [1]  **Christèle Morisseau** [1]
**Jean-Philippe Ovarlez** [1][2]  **Chengfang Ren** [2]

## Abstract

Diffusion models learn a time-indexed score field $\mathbf{s}_\theta(\mathbf{x}_t, t)$ that often inherits approximate equivariances (flips, rotations, circular shifts) from in-distribution (ID) data and convolutional backbones. Most diffusion-based out-of-distribution (OOD) detectors exploit score *magnitude* or *local geometry* (energies, curvature, covariance spectra) and largely ignore equivariances. We introduce Group-Equivariant Posterior Consistency (GEPC), a training-free probe that measures how consistently the learned score transforms under a finite group $\mathcal{G}$, detecting *equivariance breaking* even when score magnitude remains unchanged. At the population level, we propose the ideal GEPC residual, which averages an equivariance-residual functional over $\mathcal{G}$, and we derive ID upper bounds and OOD lower bounds under mild assumptions. GEPC requires only score evaluations and produces interpretable equivariance-breaking maps. On OOD image benchmark datasets, we show that GEPC achieves competitive or improved AUROC compared to recent diffusion-based baselines while remaining computationally lightweight. On high-resolution synthetic aperture radar imagery where OOD corresponds to targets or anomalies in clutter, GEPC yields strong target-background separation and visually interpretable equivariance-breaking maps. The official implementation is available at https://github.com/RouzAY/gepc-diffusion/.

[1]DEMR, ONERA, Université Paris-Saclay, 91120 Palaiseau, France [2]SONDRA, CentraleSupélec, Université Paris-Saclay, 91190 Gif-sur-Yvette, France. Correspondence to: Yadang Alexis Rouzoumka <yadang-alexis.rouzoumka@centralesupelec.fr, rouzoumkaalexis@yahoo.fr>.

*Proceedings of the 43rd International Conference on Machine Learning*, Seoul, South Korea. PMLR 306, 2026. Copyright 2026 by the author(s).

## 1. Introduction

Detecting out-of-distribution (OOD) inputs is a fundamental challenge for deploying reliable machine learning models. Classic post-hoc scores for classifiers rely on confidence or energy, such as maximum softmax probability (MSP), ODIN, and energy-based scores (Hendrycks & Gimpel, 2017; Liang et al., 2018; Liu et al., 2020), while subsequent work exploits representation geometry (e.g., $k$NN- or PCA-style feature models) (Sun et al., 2022; Guan et al., 2023).

Diffusion models (Ho et al., 2020; Song et al., 2021; Karras et al., 2022; Yang et al., 2023) have recently emerged as strong priors for OOD and anomaly detection. Beyond raw likelihoods, they expose a time-indexed score field and a generative trajectory, motivating diffusion OOD scores that often rely on either (i) trajectory/energy criteria along the reverse process or probability-flow ODE (Graham et al., 2023; Heng et al., 2024; Shin et al., 2023), or (ii) local score-field geometry such as curvature or covariance-spectrum diagnostics (Barkley et al., 2026; Shoushtari et al., 2026). These approaches primarily exploit score magnitude or local differential structure, and may require additional reverse steps or Jacobian-related computations.

In parallel, explicitly equivariant score-based and diffusion models have advanced rapidly, especially for 3D and molecular data. E(3)-equivariant diffusion models (Hoogeboom et al., 2022; Cornet et al., 2024) combine invariant noise processes with equivariant networks to guarantee that learned distributions inherit known symmetries. Recent analyses (Chen et al., 2024; Tahmasebi & Jegelka, 2024) relate score matching to a symmetrized score term plus a deviation-from-equivariance penalty, while group-convolutional / steerable CNNs (Cohen & Welling, 2016; 2017) and studies of approximate shift equivariance in vanilla CNNs (Zhang, 2019; Bruintjes et al., 2023) show that augmentation and anti-aliasing yield only approximate equivariance in practice.

These works primarily treat equivariance as an inductive bias for training. We take the complementary viewpoint: we do not enforce equivariance at training time; we measure its

(in)consistency as a test-time statistic for OOD detection.

**Our perspective: equivariance breaking as an OOD signal.** We hypothesize that when the in-distribution (ID) is approximately invariant under a group $\mathcal{G}$ (e.g., flips, rotations, circular shifts) and the backbone is convolutional and trained with augmentations, the learned diffusion scores should be *approximately $\mathcal{G}$-equivariant* on ID samples, but this *posterior consistency* should break for OOD inputs that violate the learned symmetries or lie far from the ID manifold. Concretely, group-transforming a noisy input $\mathbf{x}_t$ and transporting the predicted score back should preserve the score on ID; systematic violations indicate distribution shift. Importantly, this is not a pixel-space invariance test: we probe equivariance of the *learned score field* at noisy levels, hence the model's posterior geometry rather than raw image symmetries.

We operationalise this via **GEPC** (Group-Equivariant Posterior Consistency), a training-free probe of pretrained diffusion models. For a group $\mathcal{G}$ and an operator $\mathcal{P}_g \in \mathcal{G}$ and selected timesteps, we compare $\mathcal{P}_g^\top \mathbf{s}_\theta(\mathcal{P}_g \mathbf{x}_t, t)$ and $\mathbf{s}_\theta(\mathbf{x}_t, t)$, aggregate residuals over $\mathcal{G}$ and $t$, and calibrate the resulting statistic using only ID data. GEPC produces both a scalar OOD score and spatial heatmaps highlighting equivariance failures.

Figure 1 summarizes GEPC: we noise the input, probe score-field equivariance via group transports, aggregate residuals across timesteps, and calibrate using ID-only statistics to obtain an OOD score and equivariance-breaking maps.

**Relation to equivariance-based conformal OOD detectors (iDECODe).** iDECODe (Kaur et al., 2022) turns equivariance violations under random group actions into a conformal non-conformity score, enabling distribution-free calibrated decisions. GEPC is complementary: rather than wrapping equivariance errors in a conformal layer, we probe pretrained diffusion score fields across timesteps and analyze the corresponding *population* equivariance-breaking functional, yielding ID upper and OOD lower bounds under mild score-error assumptions.

**Relation to diffusion OOD geometry.** GEPC complements the dominant diffusion OOD families above. Trajectory/energy and curvature/covariance-spectrum methods probe the *local* geometry of $\mathbf{s}_\theta$ along time, and some require Jacobian-related computations. GEPC instead targets *global group consistency*: we measure how consistently the score transforms under $\mathcal{G}$ and turn deviations from equivariance into an OOD statistic, without computing any Jacobian or modifying the backbone.

At the population level, we give an equivariance-breaking characterization of the ideal GEPC residual under $\mathcal{G}$, closely related to deviation-from-equivariance analyses in equivariant score matching (Chen et al., 2024). Under mild assumptions, we derive ID upper bounds and OOD lower bounds for the expected GEPC residual, clarifying when posterior consistency should hold or break.

**Contributions.** (1) We introduce **GEPC**, a training-free OOD score that tests *group-consistency* of diffusion score fields across timestep and group actions. GEPC requires only inference access to a pretrained DDPM-style backbone (including improved diffusion), with no architectural changes, fine-tuning, or Jacobian evaluation.
(2) We provide a practical recipe combining group pooling, stability-based timestep selection, ID-only calibration (KDE or vector Mahalanobis), and stochastic subsampling of timestep and group elements. We characterise the computational cost and show that GEPC operates in a similar number-of-function-evaluations (NFE) regime as simple score-norm baselines while approaching the performance of more expensive trajectory and curvature-based methods.
(3) We provide a population-level analysis of GEPC: we relate the ideal residual to an equivariance-breaking functional under $\mathcal{G}$, derive ID upper bounds and OOD lower bounds under mild score-error assumptions, and discuss cross-backbone regimes where the diffusion model is trained on a different source distribution.
(4) We empirically show that GEPC is competitive with and complementary to curvature, spectrum, and trajectory-based diffusion OOD scores on CIFAR-scale near/far OOD benchmarks under a shared CelebA backbone, and that in a cross-domain high-resolution setting where a $256 \times 256$ LSUN-trained backbone is applied to radar SAR imagery, GEPC yields strong detection performance and interpretable equivariance-breaking maps.

## 2. Related Work

**OOD detection with discriminative models.** Post-hoc OOD scores for classifiers are often defined on logits or penultimate features: maximum softmax probability (MSP), ODIN, and energy-based scores (Hendrycks & Gimpel, 2017; Liang et al., 2018; Liu et al., 2020); deep $k$NN and class-aware feature decoupling further exploit representation geometry (Sun et al., 2022; Ling et al., 2025); gradient-based projections and PCA / kernel PCA probe feature manifolds (Behpour et al., 2023; Guan et al., 2023; Fang et al., 2024). A complementary line builds explicitly on *matrix-induced distances* and covariance geometry: Mahalanobis-based detectors fit a Gaussian model on ID features and use the induced distance as an OOD score (Lee et al., 2018), while residual-space methods such as ViM and NECO weight directions in the residual subspace or exploit neural-collapse structure (Wang et al., 2022; Ammar et al., 2024). Recent work further adapts the effective

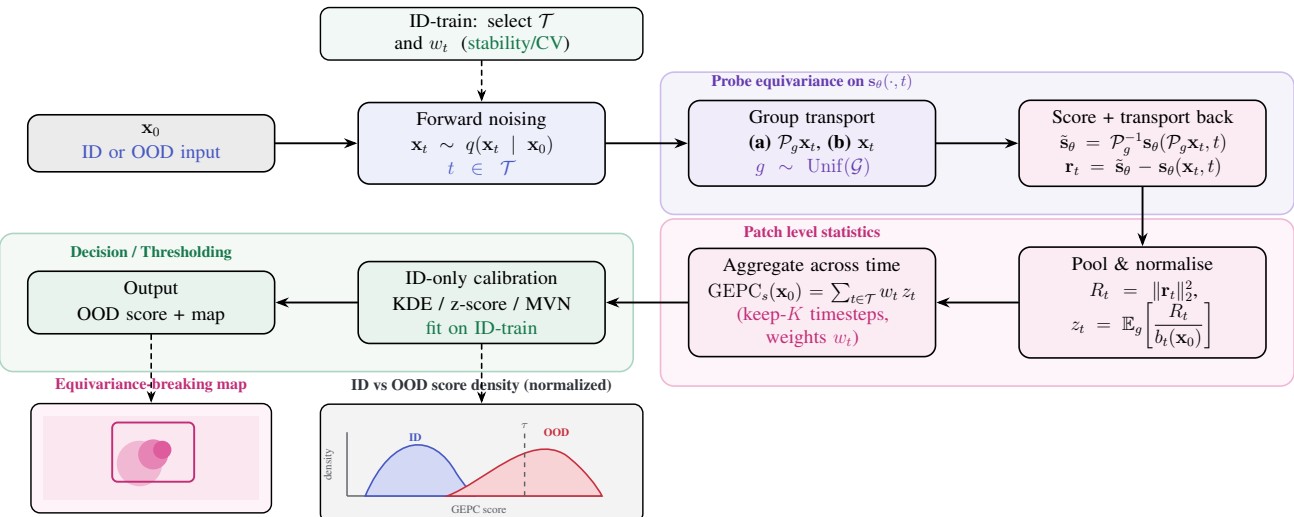

*Figure 1.* **GEPC.** We probe group-consistency of a pretrained diffusion score field by transporting $\mathbf{x}_t$ under $g \in \mathcal{G}$, transporting scores back, and measuring $\mathbf{r}_t$. Residual energies are pooled, aggregated over selected timesteps, and calibrated with ID-only statistics, yielding an OOD score and equivariance-breaking maps.

covariance at test time using the current feature, shrinking directions aligned with residual activations (Guo et al., 2025), and studies how controlling neural collapse via entropy regularization trades off OOD detection and OOD generalization (Harun et al., 2025). All these approaches operate in classifier feature space; our work is orthogonal in that we probe the *score field* of a generative model through group equivariance.

**Diffusion models for OOD and anomaly detection.** Diffusion models (Ho et al., 2020; Song et al., 2021; Karras et al., 2022; Yang et al., 2023) have been adapted to OOD via denoising- and reconstruction-based scores, trajectory energies and path discrepancies (DiffPath) (Heng et al., 2024), perturbation robustness (SPR) (Shin et al., 2023), and curvature or covariance-based diagnostics (SCOPED, EigenScore) (Barkley et al., 2026; Shoushtari et al., 2026). These methods typically exploit score magnitude or local geometry along time and often require additional reverse steps or Jacobian–vector products/power iterations. GEPC is complementary: it probes global group consistency of noised distributions via equivariance residuals, without computing Jacobian or modifying the backbone, and can be combined with curvature or trajectory-based scores.

**Equivariance and score-based models.** Equivariant score-based generative models combine group-equivariant parameterizations with score matching to model symmetric distributions efficiently (Niu et al., 2020; Cohen & Welling, 2016; 2017; Chen et al., 2024), while standard CNNs exhibit only approximate equivariance, degraded by subsampling and mitigated by anti-aliasing (Zhang, 2019; Bruintjes et al., 2023). GEPC takes a diagnostic angle: we treat group transports as a probe on a fixed pretrained diffusion model and interpret equivariance residuals as an empirical symmetry-

breaking functional that separates ID and OOD.

**Conformal and equivariance-based OOD detection.** iDE-CODe (Kaur et al., 2022) uses equivariance deviations as a conformal non-conformity score to obtain distribution-free calibrated decisions under random group actions. GEPC is not a conformal method per se, but its multi-$t$ equivariance features can, in principle, be wrapped in a conformal layer when distribution-free guarantees are required.

**Equivariance as an inductive bias for OOD detection.** Beyond score-based models, equivariance has also been used as an explicit inductive bias in discriminative unsupervised OOD detectors, e.g., via equivariant contrastive learning with soft cluster-aware semantics (Huang et al., 2025). This line is complementary to GEPC: we do not modify training or architecture, but instead use equivariance breaking of a pretrained diffusion score field as a test-time OOD signal.

## 3. Background

### 3.1. Diffusion and score-based models

We briefly review the foundations of DDPMs (Ho et al., 2020; Nichol & Dhariwal, 2021). Given data $\mathbf{x}_0 \sim q(\mathbf{x}_0)$ in $\mathbb{R}^d$, we define a forward process that generates latent variables $\mathbf{x}_1$ through $\mathbf{x}_T$ by adding a white Gaussian noise of variance $\beta_t$ at time $t$ as follows:

$$q(\mathbf{x}_t \mid \mathbf{x}_{t-1}) = \mathcal{N}\big(\mathbf{x}_t; \sqrt{\alpha_t}\mathbf{x}_{t-1}, \beta_t \mathbf{I}\big), \ t = 1, \dots, T. \quad (1)$$

where $\alpha_t = 1 - \beta_t$ with $\beta_t \in (0, 1)$. Alternatively, we can formulate the marginal at time $t$ directly as:

$$q(\mathbf{x}_t \mid \mathbf{x}_0) = \mathcal{N}\big(\mathbf{x}_t; \sqrt{\bar{\alpha}_t}\,\mathbf{x}_0, (1 - \bar{\alpha}_t)\,\mathbf{I}\big), \quad (2)$$

with $\bar{\alpha}_t = \prod_{s=1}^{t} \alpha_s$. We will slightly abuse notation and refer to the forward marginal distribution of $\mathbf{x}_t$ either as $q_t(\mathbf{x}_t)$ or simply as $q_t$ when no ambiguity arises.

Equivalently, we can sample $\mathbf{x}_t$ via the reparameterization

$$\mathbf{x}_t = \sqrt{\bar{\alpha}_t}\,\mathbf{x}_0 + \sqrt{1 - \bar{\alpha}_t}\,\boldsymbol{\epsilon}\,, \qquad (3)$$

where $\boldsymbol{\epsilon} \sim \mathcal{N}(\mathbf{0}, \mathbf{I})$ is independent of $\mathbf{x}_0$.

A generative model approximates the reverse conditionals $q(\mathbf{x}_{t-1} \mid \mathbf{x}_t)$ by Gaussian distributions $p_\theta(\mathbf{x}_{t-1} \mid \mathbf{x}_t) = \mathcal{N}(\mathbf{x}_{t-1}; \mu_\theta(\mathbf{x}_t, t), \tilde{\beta}_t \mathbf{I})$, $p_\theta(\mathbf{x}_T) = \mathcal{N}(\mathbf{0}, \mathbf{I})$ where $\tilde{\beta}_t$ is a fixed reverse variance schedule (e.g. the DDPM posterior variance). It is typically trained via the "simple" denoising objective:

$$\mathcal{L}_{\text{simple}}(\theta) = \mathbb{E}_{t,\mathbf{x}_0,\boldsymbol{\epsilon}}\big[\|\boldsymbol{\epsilon} - \boldsymbol{\epsilon}_\theta(\mathbf{x}_t, t)\|_2^2\big]\,, \qquad (4)$$

where $t$ is sampled from a fixed distribution on $\{1, \ldots, T\}$ (often uniform) and $\boldsymbol{\epsilon}_\theta(\mathbf{x}_t, t)$ denotes the noise-prediction network (e.g. a U-Net) trained to predict the forward noise $\boldsymbol{\epsilon}$ in $\mathbf{x}_t = \sqrt{\bar{\alpha}_t}\,\mathbf{x}_0 + \sigma_t\boldsymbol{\epsilon}$ with $\sigma_t^2 = 1 - \bar{\alpha}_t$. Under the MSE objective, the pointwise optimum satisfies $\boldsymbol{\epsilon}_\theta(\mathbf{x}_t, t) = \mathbb{E}[\boldsymbol{\epsilon} \mid \mathbf{x}_t]$, hence the associated score estimate is

$$\mathbf{s}_\theta(\mathbf{x}_t, t) := -\sigma_t^{-1}\,\boldsymbol{\epsilon}_\theta(\mathbf{x}_t, t)\,. \qquad (5)$$

See Appendix A.2 for a detailed derivation.

### 3.2. Scores and equivariance

For any non-degenerate distribution $p$, we denote by $\nabla_\mathbf{x}$ the (vector) gradient w.r.t. $\mathbf{x} \in \mathbb{R}^d$; thus $\nabla_\mathbf{x} \log p(\mathbf{x}) \in \mathbb{R}^d$. Let

$$\mathbf{s}_p(\mathbf{x}) := \nabla_\mathbf{x} \log p(\mathbf{x})\,, \qquad (6)$$

denote the corresponding ideal score at time $t$. Thus, for any marginal $q_t(\mathbf{x}_t)$ of the forward diffusion process used to noise the data, the estimator $\mathbf{s}_\theta$ defined in equation (5) aims to predict the corresponding deterministic score $\mathbf{s}_{q_t}$, as explained in Appendix A.2.

Let $\mathcal{G}$ be a finite group acting on $\mathbb{R}^d$ via orthogonal matrices: for any $g \in \mathcal{G}$, we denote $\mathcal{P}_g$ the corresponding operator, with $\mathcal{P}_g^\top \mathcal{P}_g = \mathbf{I}_d$.

We say a distribution $p$ on $\mathbb{R}^d$ is $\mathcal{G}$-invariant if

$$X \sim p \implies \mathcal{P}_g X \overset{d}{=} X, \qquad \forall g \in \mathcal{G}\,. \qquad (7)$$

Since each $g$ is orthogonal, then (7) is equivalent to $p(\mathcal{P}_g\mathbf{x}) = p(\mathbf{x})$. In that case, the score is $\mathcal{G}$-equivariant:

$$\mathbf{s}_p(\mathcal{P}_g\mathbf{x}) = \mathcal{P}_g\mathbf{s}_p(\mathbf{x}), \qquad \forall \mathbf{x}, \, \forall g \in \mathcal{G}\,, \qquad (8)$$

as can be seen by differentiating $\log p(\mathcal{P}_g\mathbf{x}) = \log p(\mathbf{x})$ and using $\mathcal{P}_g^\top = \mathcal{P}_g^{-1}$; see Appendix B.1. If $q_0$ is approximately

$\mathcal{G}$-invariant and the forward noise is isotropic, then each $q_t$ remains approximately $\mathcal{G}$-invariant, and the corresponding scores remain approximately $\mathcal{G}$-equivariant.

In practice, approximate equivariance arises because denoising score matching fits $\mathbf{s}_\theta(\cdot, t)$ to the ideal score $\mathbf{s}_{q_t}(\cdot, t) = \nabla_\mathbf{x} \log q_t(\mathbf{x})$ in expectation over $\mathbf{x} \sim q_t$. Indeed, the learned score $\mathbf{s}_\theta$ appears to inherit the approximate $s_{q_t}$ equivariance in high-density regions, where the training loss is concentrated. Outside these regions, the objective provides little constraint, and equivariance may be violated arbitrarily. Architectural biases such as translation-equivariant convolutions and data augmentation can further promote such approximate symmetries. In cross-backbone settings, however, this learned equivariance is not expected to persist far from the source high-density region, which motivates the distance-to-manifold perspective in Section 4.

## 4. GEPC: Group-Equivariant Posterior Consistency

For any vector field $f(\cdot, t)$ and any $g \in \mathcal{G}$ acting on $\mathbb{R}^d$ through an orthogonal matrix $\mathcal{P}_g$ (so $\mathcal{P}_g^{-1} = \mathcal{P}_g^\top$), define the equivariance residual operator

$$\Delta_g f(\mathbf{x}, t) := \mathcal{P}_g^{-1} f(\mathcal{P}_g\mathbf{x}, t) - f(\mathbf{x}, t)\,. \qquad (9)$$

**Definition 4.1** (GEPC). Let $\mathbf{s}_\theta(\cdot, t)$ denote the score field of a pretrained diffusion backbone. Given an input $\mathbf{x}_0$, sample $\mathbf{x}_t \sim q(\mathbf{x}_t \mid \mathbf{x}_0)$ from the forward noising process. Define the equivariance residual

$$R_t(\mathbf{x}_t, g) := \|\Delta_g \mathbf{s}_\theta(\mathbf{x}_t, t)\|_2^2\,, \qquad (10)$$

and the GEPC score

$$\text{GEPC}(\mathbf{x}_0) := \sum_{t \in \mathcal{T}} w_t \mathbb{E}_{\mathbf{x}_t \sim q(\cdot|\mathbf{x}_0),\, g \sim \nu_\mathcal{G}}\big[R_t(\mathbf{x}_t, g)\big]\,, \quad (11)$$

where $\nu_\mathcal{G}$ is uniform over the finite set $\mathcal{G}$, and $w_t \geq 0$ with $\sum_{t \in \mathcal{T}} w_t = 1$.

**Why equivariance, not $\|\mathbf{s}_\theta\|$? (Gaussian mean-shift).** Let $p = \mathcal{N}(\mu, \sigma^2 \mathbf{I}_d)$, whose score is $\mathbf{s}(\mathbf{x}) = -(\mathbf{x} - \mu)/\sigma^2$. Then $\mathbb{E}_{\mathbf{x} \sim p}\big[\|\mathbf{s}(\mathbf{x})\|_2^2\big] = d/\sigma^2$ ($d$ being the dimension of $\mathbf{x}$), independent of $\mu$. In contrast, the equivariance residual detects mean shifts. For $\mathcal{G} = \{\mathbf{I}_d, -\mathbf{I}_d\}$ with uniform $\nu_\mathcal{G}$,

$$\mathbb{E}_{g \sim \nu_\mathcal{G}}\big[\|\Delta_g \mathbf{s}(\mathbf{x})\|_2^2\big] = \frac{2}{\sigma^4}\|\mu\|_2^2, \qquad (12)$$

which separates $\mu = \mathbf{0}$ (centered / invariant) from $\mu \neq \mathbf{0}$ (non-invariant), even though $\|\mathbf{s}(\mathbf{x})\|$ does not. Further checks are in Appendix C. This intuition from the Gaussian example is confirmed in Figure 4, where GEPC shows better separation than $\|\mathbf{s}_\theta(\mathbf{x})\|$ on real image datasets.

**Decomposition.** Fix a time $t$ and let $p_t$ be any test marginal density of $\mathbf{x}_t$. Its ideal score is $\mathbf{s}_{p_t}(\mathbf{x}) := \nabla_{\mathbf{x}} \log p_t(\mathbf{x})$, and the score approximation error is

$$\mathbf{e}_{p_t}(\mathbf{x}, t) := \mathbf{s}_\theta(\mathbf{x}, t) - \mathbf{s}_{p_t}(\mathbf{x}). \tag{13}$$

Define the equivariance-breaking functional

$$\mathcal{B}^{(\mathcal{G})}(p_t) := \mathbb{E}_{\mathbf{x} \sim p_t,\, g \sim \nu_{\mathcal{G}}} \left[ \|\Delta_g \mathbf{s}_{p_t}(\mathbf{x}, t)\|_2^2 \right]. \tag{14}$$

If $p_t$ is a $\mathcal{G}$-invariant distribution, then $\mathcal{B}^{(\mathcal{G})}(p_t) = 0$ since invariance is equivalent to score equivariance (Appendix B.1).

**Expected residual bounds (ID vs OOD).** Let $q_t$ denote the time-$t$ marginal distribution induced by the ID training distribution $q(\mathbf{x}_0)$, and let $p_t$ denote the time-$t$ marginal distribution induced by any test distribution.

**Proposition 4.2** (Expected GEPC residual bounds). *For any marginal $p_t$, define*

$$\Delta_E(p_t, t) := \mathbb{E}_{\mathbf{x} \sim p_t,\, g \sim \nu_{\mathcal{G}}} \left[ \|\mathbf{e}_{p_t}(\mathcal{P}_g \mathbf{x}, t) - \mathbf{e}_{p_t}(\mathbf{x}, t)\|_2^2 \right].$$

*With the shorthand $\mathbb{E}_{p_t, g}[\cdot] := \mathbb{E}_{\mathbf{x} \sim p_t,\, g \sim \nu_{\mathcal{G}}}[\cdot]$, we have*

$$\mathbb{E}_{p_t, g}\left[ R_t(\mathbf{x}, g) \right] \leq 2\,\mathcal{B}^{(\mathcal{G})}(p_t) + 4\,\mathbb{E}_{\mathbf{x} \sim p_t}\left[ \|\mathbf{e}_{p_t}(\mathbf{x}, t)\|_2^2 \right]$$
$$+ 4\,\mathbb{E}_{p_t, g}\left[ \|\mathbf{e}_{p_t}(\mathcal{P}_g \mathbf{x}, t)\|_2^2 \right] := u_b(p_t),$$

$$\mathbb{E}_{p_t, g}\left[ R_t(\mathbf{x}, g) \right] \geq \mathcal{B}^{(\mathcal{G})}(p_t) + \Delta_E(p_t, t) \tag{15}$$
$$- 2\sqrt{\mathcal{B}^{(\mathcal{G})}(p_t)\, \Delta_E(p_t, t)} := l_b(p_t).$$

The proof is provided in Appendix B.2.

**Backbone trained on ID.** In the ideal detection regime, the ID expected residual is small while the OOD expected residual is large: $u_b(q_t) \ll l_b(p_t)$ for relevant OOD marginals $p_t$. When the backbone is well trained on $q_t$, the score error $\mathbb{E}_{\mathbf{x} \sim q_t} \|\mathbf{e}_{q_t}(\mathbf{x}, t)\|_2^2$ is small. Moreover, $\mathbb{E}_{\mathbf{x} \sim q_t,\, g} \|\mathbf{e}_{q_t}(\mathcal{P}_g \mathbf{x}, t)\|_2^2$ remains small if the backbone preserves score consistency under $\mathcal{G}$ transformations, often observed for convolutional architectures on approximately invariant data (Section 3.2). Finally, when $q_t$ is approximately $\mathcal{G}$-invariant, $\mathcal{B}^{(\mathcal{G})}(q_t)$ is also small, so $u_b(q_t)$ is small. For an OOD marginal $p_t$ that violates the assumed invariances, $\mathcal{B}^{(\mathcal{G})}(p_t)$ and/or the error terms increase, pushing $l_b(p_t)$ upward, which formalizes how GEPC separates ID from OOD via non-invariance and score mismatch.

**Cross-backbone case.** In cross-backbone detection, the backbone is trained on a *source* distribution $r(\mathbf{x}_0)$ while detection is performed on another ID distribution $q(\mathbf{x}_0)$ (and OODs). Score accuracy is then expected only near high-density regions under the source marginal $r_t$. We model this by an effective source manifold of $r_t$, $\mathcal{M}_t$, and the ambient space $\mathcal{N}_t$ of a distribution $p_t$ such that $\mathcal{N}_t \supset \mathcal{M}_t$.

We denote the projection $\pi_t : \mathcal{N}_t \to \mathcal{M}_t$ commuting with the group action. Define $d_t(\mathbf{x}) := \|\mathbf{x} - \pi_t(\mathbf{x})\|_2$ and assume $\mathbf{s}_\theta(\cdot, t)$ is $L_t$-Lipschitz on $\mathcal{N}_t$:

$$\|\mathbf{s}_\theta(\mathbf{x}, t) - \mathbf{s}_\theta(\mathbf{y}, t)\|_2 \leq L_t \|\mathbf{x} - \mathbf{y}\|_2, \quad \forall \mathbf{x}, \mathbf{y} \in \mathcal{N}_t. \tag{16}$$

**Proposition 4.3** (Cross-backbone pointwise bounds). *Assume (16) and $\pi_t(\mathcal{P}_g \mathbf{x}) = \mathcal{P}_g \pi_t(\mathbf{x})$ for all $\mathbf{x} \in \mathcal{N}_t$, $g \in \mathcal{G}$. Then, for any $\mathbf{x} \in \mathcal{N}_t$,*

$$\mathbb{E}_{g \sim \nu_{\mathcal{G}}} \left[ R_t(\mathbf{x}, g) \right] \leq 2\,\mathbb{E}_{g \sim \nu_{\mathcal{G}}} \left[ R_t(\pi_t(\mathbf{x}), g) \right] + 8 L_t^2\, d_t(\mathbf{x})^2. \tag{17}$$

*If moreover there exist $m_t > 0$ and $d_{0,t} \geq 0$ such that for all $\mathbf{x} \in \mathcal{N}_t$ with $d_t(\mathbf{x}) \geq d_{0,t}$,*

$$\left\langle \mathbf{s}_\theta(\mathbf{x}, t) - \mathbf{s}_\theta(\pi_t(\mathbf{x}), t), \frac{\mathbf{x} - \pi_t(\mathbf{x})}{\|\mathbf{x} - \pi_t(\mathbf{x})\|_2} \right\rangle \leq -m_t\, d_t(\mathbf{x}), \tag{18}$$

*then, writing $\rho_t(\mathbf{x}) := \sqrt{\mathbb{E}_{g \sim \nu_{\mathcal{G}}} \left[ R_t(\pi_t(\mathbf{x}), g) \right]}$, we have*

$$\mathbb{E}_{g \sim \nu_{\mathcal{G}}} \left[ R_t(\mathbf{x}, g) \right] \geq \left( \left( (m_t - L_t)\, d_t(\mathbf{x}) - \rho_t(\mathbf{x}) \right)_+ \right)^2, \tag{19}$$

*where $(a)_+ := \max\{a, 0\}$.*

The proof is provided in Appendix B.3, and the derivation of the regularity hypotheses is discussed in Appendix A.3.

**Implications for detection.** If the backbone is accurate and approximately equivariant on the high-density region of the source distribution, we may assume that $\mathbb{E}_g \left[ R_t(\mathbf{z}, g) \right]$ is small for $\mathbf{z} \in \mathcal{M}_t$. In this regime, the residual terms in Proposition 4.3 become negligible and the bounds are dominated by the distance-to-manifold terms (quadratic in $d_t(\mathbf{x})$), implying that the GEPC score increases as samples move away from the source manifold.

Taking expectations over $\mathbf{x} \sim p_t$ yields a comparison between in-distribution and out-of-distribution residuals: in-distribution samples satisfy an upper bound of order $8 L_t^2 \mathbb{E}_{q_t}[d_t(\mathbf{x})^2]$, whereas out-of-distribution samples exceed $(m_t - L_t)^2 \mathbb{E}_{p_t}[d_t(\mathbf{x})^2]$. This separation suggests good detection performance when $\frac{\mathbb{E}_{q_t}[d_t(\mathbf{x})^2]}{\mathbb{E}_{p_t}[d_t(\mathbf{x})^2]} \ll \left( \frac{m_t}{L_t} - 1 \right)^2$ up to constants.

## 5. Practical GEPC for DDPM

We now describe how GEPC is computed in practice for discrete-time DDPM or improved-diffusion backbones.

### 5.1. Per-sample GEPC, pooling, and normalisation

Let $\mathcal{G}$ be a set of invertible image transformations with known inverses. Throughout, unless stated otherwise, $\mathcal{G} = \{\mathrm{id}, \mathrm{flip}_x, \mathrm{flip}_y, \mathrm{rot}_{90}, \mathrm{rot}_{180}, \mathrm{shift}_x, \mathrm{shift}_y\}$ with 1-pixel circular shifts, so $|\mathcal{G}| = 7$ on $32 \times 32$ square images.

**Pooling convention.** Given a field $A \in \mathbb{R}^{C \times h \times w}$, $\mathrm{pool}(A)$ denotes a standard spatial pooling that first averages across channels and then aggregates over spatial locations by either mean-pooling or top-$k$ pooling (top-$k$ averages the $k$ largest spatial responses). With a slight abuse of notation, $\mathrm{pool}(\|.\|_2^2)$ denotes pooling applied to the pointwise squared $\ell_2$-norm over channels.

Given an input $\mathbf{x}_0$ and timestep $t$, we sample $\mathbf{x}_t$ via $\mathbf{x}_t = \sqrt{\bar{\alpha}_t}\,\mathbf{x}_0 + \sqrt{1 - \bar{\alpha}_t}\,\boldsymbol{\epsilon},\ \boldsymbol{\epsilon} \sim \mathcal{N}(\mathbf{0}, \mathbf{I})$. Define the transported score residual field

$$\mathbf{r}_t(\mathbf{x}_t, g) \coloneqq \mathcal{P}_g^{-1}\,\mathbf{s}_\theta(\mathcal{P}_g \mathbf{x}_t, t) - \mathbf{s}_\theta(\mathbf{x}_t, t) \in \mathbb{R}^{C \times h \times w}. \tag{20}$$

We also define the pooled score-energy normaliser

$$b_t(\mathbf{x}_0) \coloneqq \mathrm{pool}\big(\|\mathbf{s}_\theta(\mathbf{x}_t, t)\|_2^2\big). \tag{21}$$

Our default per-timestep GEPC scalar (denoted GEPC$_s$ in the code) is the base-normalised residual energy

$$z_t^{(s)}(\mathbf{x}_0) \coloneqq \mathbb{E}_{g \sim \mathrm{Unif}(\mathcal{G})}\big[b_t^{-1}(\mathbf{x}_0)\mathrm{pool}\big(\|\mathbf{r}_t(\mathbf{x}_t, g)\|_2^2\big)\big]. \tag{22}$$

We optionally average (22) over $m$ Monte Carlo noise draws $\boldsymbol{\epsilon}$ (`mc_samples`). Using the same transported scores $\{\mathcal{P}_g^{-1}\mathbf{s}_\theta(\mathcal{P}_g \mathbf{x}_t, t)\}_{g \in \mathcal{G}}$, we also compute alternative GEPC features, including cosine consistency, pairwise dispersion, $\mathbf{x}_0$-consistency, and cycle consistency; see Appendix E. All quadratic (L2-type) features are reported in base-normalised form (with a feature-specific normaliser when appropriate), while the cosine feature is scale-invariant and therefore left unnormalised.

Finally, we aggregate across a small set of selected timesteps $\mathcal{T}$ using `agg_t` (default: weighted mean)

$$\widehat{\mathrm{GEPC}}(\mathbf{x}_0) \coloneqq \sum_{t \in \mathcal{T}} w_t\, z_t^{(s)}(\mathbf{x}_0), \qquad \sum_{t \in \mathcal{T}} w_t = 1. \tag{23}$$

### 5.2. ID-only timestep selection and calibration

To avoid OOD-labelled tuning, we select timesteps, per-timestep weights, and calibration using ID samples only. We first form a candidate set $\mathcal{T}_{\mathrm{cand}}$ by mapping a fixed list of target schedule levels `snr_levels` to discrete indices (for DDPM schedules this is implemented by nearest-neighbour matching on $\sqrt{\bar{\alpha}_t}$).

On ID-train, for each $t \in \mathcal{T}_{\mathrm{cand}}$ we compute a stability score via the coefficient of variation, $\mathrm{CV}(t) = \frac{\mathrm{std}(u_t(\mathbf{x}))}{|\mathrm{mean}(u_t(\mathbf{x}))|}$,

where $u_t(\mathbf{x})$ is a base GEPC statistic at timestep $t$ (default: $z_t^{(s)}$). We keep the $K$ most stable steps (lowest CV), yielding $\mathcal{T}$ with $|\mathcal{T}| = K$ (`keep_k`). Optionally, we set weights $w_t \propto 1/\mathrm{CV}(t)$ and normalise them (`weight_t=inv_cv`); otherwise $w_t$ is uniform (`weight_t=none`).

**Calibration modes (ID-only).** Let $z_{t,f}(\mathbf{x})$ denote the enabled feature scalars (each OOD-high by construction). We support three ID-only calibration modes: (i) **KDE** (`density_mode=kde`): fit a 1D KDE $p_{t,f}$ per $(t, f)$ (Silverman rule-of-thumb with robust IQR bandwidth) and aggregate log-densities; (ii) **z-score** (`density_mode=zscore`): fit $(\mu_{t,f}, \sigma_{t,f})$ and use the Gaussian log-score $-\frac{1}{2}((z - \mu)/\sigma)^2$; (iii) **raw** (`density_mode=none`): no density model is fit and we directly aggregate raw OOD-high feature values. Alternatively, **vector MVN** (`vector_mode=mvn`) fits a single Gaussian/Mahalanobis model on the concatenated multi-$(t, f)$ feature vector. For all density-based modes, the final anomaly score is the negative ID score (OOD-high), matching the implementation.

### 5.3. Metrics and compute (F+J)

We report AUROC and forward-equivalent compute as $F + J$, where $F$ is one score-network forward evaluation and $J$ is one Jacobian–vector product (JVP), each counted as a forward-equivalent operation.

GEPC is fully test-time and uses only score-network evaluations. For GEPC, at each timestep $t$, we compute one reference score $\mathbf{s}_\theta(\mathbf{x}_t, t)$ and one batched evaluation over $\{\mathcal{P}_g \mathbf{x}_t\}_{g \in \mathcal{G}}$, hence $F = (1 + |\mathcal{G}|)\,|\mathcal{T}|\,m$ and $J = 0$. All GEPC feature variants reuse the same score evaluations at each $(t, g)$, so enabling additional features or feature fusion does not change $F + J$.

For methods that require a reverse trajectory of $T$ steps, we count $F = T$ score evaluations (and the corresponding $J$ terms when applicable).

## 6. Experiments

We evaluate GEPC as a diffusion-based OOD detector under two regimes: (i) CIFAR-scale benchmarks at $32 \times 32$, where we report both prior diffusion-OOD baselines and a strictly shared-backbone comparison using a single CelebA-trained improved-diffusion backbone; and (ii) a cross-domain, high-resolution setting, where a $256 \times 256$ LSUN-trained backbone is evaluated on radar SAR imagery, with OOD samples corresponding to targets or anomalies embedded in clutter. We address two questions: (i) whether GEPC is competitive with recent diffusion-based OOD scores, including under a strictly shared-backbone protocol; and (ii) whether GEPC provides robust and interpretable OOD signals when a high-resolution LSUN-trained backbone is applied cross-domain to SAR imagery.

### 6.1. Setup

**Backbones and evaluation regime.** At $32 \times 32$, our primary evaluation combines two comparison regimes. First,

we report representative diffusion-OOD baseline numbers from prior benchmarks under their original protocols when available. Second, for a strictly shared-backbone comparison, we evaluate GEPC together with shared-backbone diffusion methods using a *single* unconditional improved-diffusion backbone trained on CelebA at $32 \times 32$ using the public `improved-diffusion` codebase (Nichol & Dhariwal, 2021). This checkpoint is never fine-tuned. For high-resolution cross-domain evaluation, we further probe an unconditional LSUN-256 improved-diffusion backbone on $256 \times 256$ SAR patches.

**Baselines.** We compare GEPC against two broad classes of OOD detection methods. First, we consider **ID-trained** discriminative and generative baselines, including energy-based models such as IGEBM (Du & Mordatch, 2019), VAEBM (Xiao et al., 2021), and Improved Contrastive Divergence (CD) (Du et al., 2021), as well as Input Complexity (IC) (Serrà et al., 2020), Density of States (DOS) (Morningstar et al., 2020), Watanabe–Akaike Information Criterion (WAIC) (Choi et al., 2019), the Typicality Test (TT) (Nalisnick et al., 2019), and the Likelihood Ratio (LR) (Ren et al., 2019).

Second, we consider **diffusion-based** OOD scores. In the main table, we distinguish (i) prior diffusion baselines reported from recent benchmark papers under their original or mixed evaluation protocols, namely NLL, IC (diffusion), MSMA (Mahmood et al., 2021), DDPM-OOD (Graham et al., 2023), and LMD (Liu et al., 2023); and (ii) methods evaluated under a **shared frozen CelebA-32 backbone**, namely DiffPath (Heng et al., 2024), SCOPED (Barkley et al., 2026), and GEPC. In an additional shared-backbone comparison, we further recompute selected diffusion baselines under the same CelebA-32 checkpoint to enable a stricter apples-to-apples comparison, details regarding implementation are provided in C.3.

### 6.2. CIFAR-10 / SVHN / CelebA at $32 \times 32$

We evaluate GEPC on the low-resolution regime with three ID datasets: CIFAR-10 (C10), SVHN, and CelebA (downsampled to $32 \times 32$). To enable direct comparison with recent diffusion-OOD benchmarks under the same backbone, we report the 9 canonical ID/OOD pairs used in SCOPED (Barkley et al., 2026) and DiffPath (Heng et al., 2024).

Table 1 reports AUROC for all 9 ID/OOD pairs. The upper block groups *ID-trained* likelihood/energy-based model (EBM)-style baselines from prior work (trained per ID dataset). The middle block groups *prior diffusion-based* baselines as reported in recent benchmark papers under their original or mixed protocols. The lower block groups *shared-backbone diffusion methods* evaluated on a single pretrained CelebA improved-diffusion backbone, namely DiffPath, SCOPED, and our GEPC.

For the shared-backbone table, we re-evaluate NLL, IC (diffusion), MSMA (Mahmood et al., 2021), DDPM-OOD (Graham et al., 2023), and LMD (Liu et al., 2023) under the same frozen CelebA-32 checkpoint; DiffPath (Heng et al., 2024), SCOPED (Barkley et al., 2026), and GEPC are reported in the same protocol. Motivated by the group-ablation results in Appendix 6, we instantiate GEPC_lite as the strongest low-cost variant under the $8F$ budget, namely the flips-only subgroup. Complementary metrics to AUROC are reported in Appendix F.5.

### 6.3. Radar SAR OOD detection and localisation

We evaluate GEPC for ship/wake detection and localisation on high-resolution SAR imagery (HRSID). We define an OOD task in which *sea-clutter-only* patches are in-distribution (ID), whereas patches containing at least one ship or wake are out-of-distribution (OOD). We apply a pretrained LSUN-256 diffusion backbone *as-is* to $256 \times 256$ SAR patches, without any SAR fine-tuning, and compute GEPC patch-wise. In addition to qualitative residual maps, we report patch-level detection metrics and compare against a classical Reed–Xiaoli (RX) detector (Reed & Yu, 1990) baseline fitted on clutter-only ID training patches. GEPC residual maps remain low on homogeneous sea clutter while concentrating on ships and wakes, yielding interpretable symmetry-breaking localisation (Figure 2). Additional quantitative results, the RX baseline comparison, the SSDD cross-dataset evaluation, and further qualitative examples are provided in Appendix G and Figure 5.

### 6.4. Ablations and runtime

We conduct ablations to assess robustness, sensitivity to design choices, calibration strategies, stochastic stability, and computational cost. Detailed ablation tables across the 9 ID/OOD pairs are reported in Appendix F, along with representative plots, score histograms, timestep-selection sweeps, and Monte Carlo noise sensitivity analyses.

**Group elements.** Using our default transport set (identity, flips, rotations, and 1-pixel circular shifts), we report a diagnostic AUROC obtained by isolating each transform contribution on the *raw transported-gap component* (no KDE/z-score calibration), and compare it to the same component averaged over $\mathcal{G}$. Across pairs, performance is not dominated by a single element, supporting that GEPC captures a stable symmetry-breaking effect rather than an isolated artifact (Appendix F, Table 10 and Figure 3).

**Timestep selection and weighting.** Single-timestep AUROC-vs-$t$ curves are shown for the *raw transported-gap component* to localise where symmetry-breaking arises. Our ID-only coefficient-of-variation (CV) rule then selects a small retained set $\mathcal{T}$ (fixed $K$ across datasets for comparable compute) and achieves performance close to the

*Table 1.* AUROC for in-distribution vs. out-of-distribution tasks at $32 \times 32$ (9 standard ID/OOD pairs). Higher is better. We report compute as $F + J$ (forward passes + JVPs). The table combines: (i) ID-trained baselines from prior work; (ii) prior diffusion-based baselines reported under their original or mixed protocols; and (iii) shared-backbone diffusion methods evaluated under a single frozen CelebA-32 checkpoint.

| Method | CIFAR-10 (ID) | | | SVHN (ID) | | | CelebA (ID) | | | Avg. | $F + J$ |
|---|---|---|---|---|---|---|---|---|---|---|---|
| | SVHN | CelebA | C100 | C10 | CelebA | C100 | C10 | SVHN | C100 | | |
| *ID-trained baselines (trained per ID)* | | | | | | | | | | | |
| IC | 0.950 | 0.863 | 0.736 | – | – | – | – | – | – | – | – |
| IGEBM | 0.630 | 0.700 | 0.500 | – | – | – | – | – | – | – | – |
| VAEBM | 0.830 | 0.770 | 0.620 | – | – | – | – | – | – | – | – |
| Improved CD | 0.910 | – | **0.830** | – | – | – | – | – | – | – | – |
| DoS | 0.955 | 0.995 | 0.571 | 0.962 | **1.00** | 0.965 | 0.949 | 0.997 | 0.956 | **0.928** | – |
| WAIC | 0.143 | 0.928 | 0.532 | 0.802 | 0.991 | 0.831 | 0.507 | 0.139 | 0.535 | 0.601 | – |
| TT | 0.870 | 0.848 | 0.548 | 0.970 | **1.00** | 0.965 | 0.634 | 0.982 | 0.671 | 0.832 | – |
| LR | 0.064 | 0.914 | 0.520 | 0.819 | 0.912 | 0.779 | 0.323 | 0.028 | 0.357 | 0.524 | – |
| *Prior diffusion-based baselines (generally id-trained / mixed protocols)* | | | | | | | | | | | |
| NLL | 0.091 | 0.574 | 0.521 | **0.990** | 0.999 | **0.992** | 0.814 | 0.105 | 0.786 | 0.652 | $1000F + 0J$ |
| IC (diffusion) | 0.921 | 0.516 | 0.519 | 0.080 | 0.028 | 0.100 | 0.485 | 0.972 | 0.510 | 0.459 | $1000F + 0J$ |
| MSMA | 0.957 | **1.00** | 0.615 | 0.976 | 0.995 | 0.980 | 0.910 | 0.996 | 0.927 | **0.928** | $10F + 0J$ |
| DDPM-OOD | 0.390 | 0.659 | 0.536 | 0.951 | 0.986 | 0.945 | 0.795 | 0.636 | 0.778 | 0.742 | $350F + 0J$ |
| LMD | **0.992** | 0.557 | 0.604 | 0.919 | 0.890 | 0.881 | 0.989 | **1.00** | 0.979 | 0.868 | $10^4F + 0J$ |
| *Shared-backbone diffusion methods (single frozen CelebA-32 backbone)* | | | | | | | | | | | |
| DiffPath | 0.910 | 0.897 | 0.590 | 0.939 | 0.979 | 0.953 | 0.998 | **1.00** | 0.998 | 0.918 | $10F + 0J$ |
| SCOPED | 0.814 | 0.940 | 0.477 | 0.971 | 0.996 | 0.959 | 0.925 | 0.994 | 0.962 | 0.892 | $2F + 2J$ |
| **GEPC (ours)** | 0.842 | 0.999 | 0.554 | 0.880 | **1.00** | 0.897 | **1.00** | **1.00** | **1.00** | 0.908 | $16F + 0J$ |

In this table, non-GEPC baseline numbers are taken from prior benchmark papers. For the $32 \times 32$ pairwise tasks, most baseline rows follow DiffPath (Heng et al., 2024) and SCOPED (Barkley et al., 2026); WAIC/TT/LR rows are as reported in (Morningstar et al., 2020).

best single-timestep choices without any OOD labels (Appendix F, Table 4 and Figure 3).

**Calibration and feature fusion.** We compare 1D KDE calibration to z-score normalisation, the raw uncalibrated score, and a Gaussian/Mahalanobis model on multi-$t$ GEPC feature vectors. We also ablate feature fusion via mean (Table 12).

**Runtime and NFEs.** GEPC requires no backpropagation, Jacobian-vector products, nor fine-tuning. For each retained timestep $t$, we evaluate one reference score field $\mathbf{s}_\theta(\mathbf{x}_t, t)$ and one batched evaluation over transported inputs $\{\mathcal{P}_g \mathbf{x}_t\}_{g \in \mathcal{G}}$, i.e. $(1 + |\mathcal{G}|)$ forward passes per timestep. With $m$ Monte-Carlo noise samples and $K = |\mathcal{T}|$ retained timesteps, the total cost is NFE $= (1 + |\mathcal{G}|)Km$ forward passes per input, parallelisable over $g$ (and, memory permitting, over $t$). We report the accuracy-compute trade-off via a sweep over $K$ with the implied NFE in Appendix F.2 (Table 4), and provide measured wall-clock timing in Appendix F.10.

**Representative plots.** For readability, we visualise per-transform and per-timestep behaviours on a representative pair (SVHN as ID, CIFAR-100 as OOD) in Figure 3, and show score histograms in Figure 4. Complete 9-pair ablation tables are provided in Appendix F.

## 7. Conclusion and discussions

GEPC enables OOD detection with diffusion models by leveraging symmetry properties. It achieves competitive performance both with an ID-trained backbone and in a training-free ID setting, and provides equivariance maps that facilitate detection on complex images such as SAR imagery.

**Computational cost.** GEPC requires multiple score evaluations per input; stochastic subsampling of group elements and timestep reduces cost but remains higher than scalar diagnostics such as score norm. However, GEPC avoids Jacobian/Hessian evaluations and remains competitive in NFE with many diffusion-based baselines.

**Symmetry assumptions.** GEPC relies on approximate invariances under a chosen group $\mathcal{G}$. For modalities lacking such symmetries (e.g., strongly oriented or structured data), performance may degrade or require adapting $\mathcal{G}$ (e.g., using learned or domain-specific transformations).

**Backbone reliance.** GEPC requires a pretrained diffusion backbone, which may not be available for all domains. Our cross-backbone experiments, however, suggest that even mismatched backbones can be informative, consistent with recent "foundation" diffusion models reused across tasks.

**Relation to dynamic covariance calibration and neural-**

*Table 2.* Shared-backbone comparison on a single frozen CelebA-32 improved-diffusion checkpoint. AUROC for the 9 standard $32 \times 32$ ID/OOD pairs. Higher is better. All methods below are evaluated under the same frozen backbone; they differ only by their test-time statistic or reconstruction rule. We report compute as $F + J$ (forward passes + JVPs).

| | CIFAR-10 (ID) | | | SVHN (ID) | | | CelebA (ID) | | | Avg. | $F + J$ |
|---|---|---|---|---|---|---|---|---|---|---|---|
| Method | SVHN | CelebA | C100 | C10 | CelebA | C100 | C10 | SVHN | C100 | | |
| NLL | 0.066 | 0.180 | 0.506 | 0.933 | 0.859 | 0.924 | 0.822 | 0.141 | 0.798 | 0.581 | $4000F + 0J$ |
| IC (diffusion) | 0.703 | 0.072 | 0.549 | 0.296 | 0.023 | 0.343 | 0.929 | 0.977 | 0.935 | 0.536 | $4000F + 0J$ |
| MSMA | 0.767 | 0.818 | 0.534 | 0.884 | 0.960 | 0.892 | 0.952 | 0.991 | 0.948 | 0.861 | $10F + 0J$ |
| DDPM-OOD | 0.076 | 0.114 | 0.473 | 0.936 | 0.697 | 0.915 | 0.895 | 0.310 | 0.861 | 0.586 | $364F + 0J$ |
| LMD | 0.366 | 0.713 | 0.572 | 0.634 | 0.778 | 0.681 | 0.284 | 0.223 | 0.385 | 0.515 | $10^4F + 0J$ |
| DiffPath | **0.910** | 0.897 | **0.590** | 0.939 | 0.979 | 0.953 | 0.998 | **1.000** | 0.998 | **0.918** | $10F + 0J$ |
| SCOPED | 0.814 | 0.940 | 0.477 | **0.971** | 0.996 | **0.959** | 0.925 | 0.994 | 0.962 | 0.892 | $2F + 2J$ |
| **GEPC_lite (ours)** | 0.821 | 0.990 | 0.548 | 0.906 | **1.00** | 0.928 | **1.00** | **1.00** | 0.999 | 0.910 | $8F + 0J$ |
| **GEPC (ours)** | 0.842 | **0.999** | 0.554 | 0.880 | **1.00** | 0.897 | **1.00** | **1.00** | **1.00** | 0.908 | $16F + 0J$ |

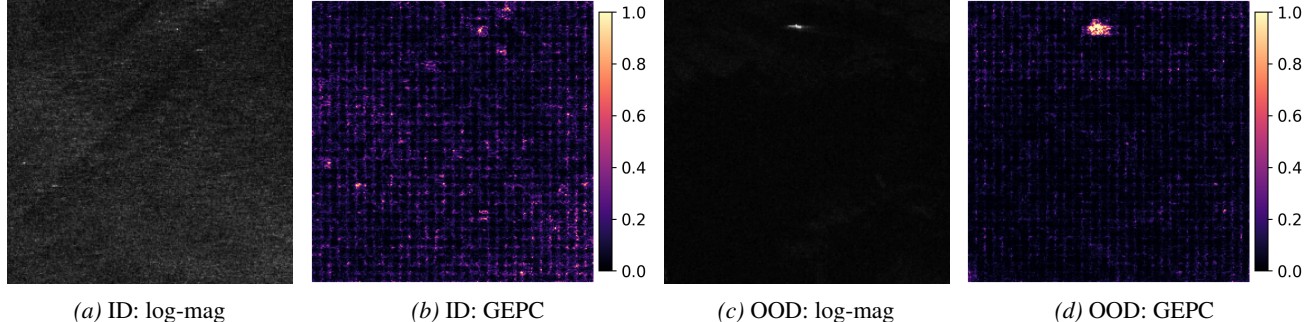

*(a)* ID: log-mag      *(b)* ID: GEPC      *(c)* OOD: log-mag      *(d)* OOD: GEPC

*Figure 2.* GEPC on HRSID SAR imagery (LSUN-256 backbone, no SAR fine-tuning). We visualise the pre-pooling residual magnitude map using a *global* normalisation (shared scale) to enable direct comparison between ID and OOD (Appendix G, Figure 5).

**collapse-based OOD.** Feature-space approaches that adapt covariance geometry or exploit neural collapse structure (Guo et al., 2025; Ammar et al., 2024; Harun et al., 2025) are complementary to GEPC: they refine matrix-induced distances on classifier features, whereas GEPC probes equivariance breaking directly in the diffusion score field.

**Extensions.** Future work includes extending GEPC beyond hand-specified finite groups, e.g., through continuous or steerable group actions and ID-validated domain-specific transformations. Another promising direction is to apply the same transport-consistency principle in learned representation spaces, which could improve scalability and enable closer comparisons with large-scale OOD benchmarks and post-hoc feature-based detectors. Finally, GEPC could be studied with non-convolutional diffusion backbones such as DiT, combined with curvature and path-based diagnostics, and extended to multi-modal diffusion models.

## Acknowledgements

Part of this work was supported by ANR-ASTRID NEPTUNE 3 (ANR-23-ASM2-0009).

## Impact Statement

This paper advances out-of-distribution detection for diffusion models, with potential applications in safety-critical sensing scenarios such as anomaly detection in radar imaging; we do not anticipate specific negative societal impacts beyond standard considerations in machine learning.

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

# A. Diffusion and score-matching identities (detailed)

We collect detailed derivations for the identities used in the main text: (i) denoising-score identities linking $\epsilon$-prediction to scores, (ii) Tweedie's formula under DDPM scaling, (iii) posterior covariance identities and their relation to the Jacobian, (iv) Lipschitz / contractivity properties derived from posterior covariance bounds.

## A.1. Forward noising closed-form (DDPM)

Recall that the forward diffusion process is defined by

$$q(\mathbf{x}_t \mid \mathbf{x}_{t-1}) = \mathcal{N}\left(\mathbf{x}_t; \sqrt{\alpha_t}\,\mathbf{x}_{t-1}, \beta_t\,\mathbf{I}\right), \qquad \alpha_t = 1 - \beta_t, \tag{24}$$

where we denote $\bar{\alpha}_t = \prod_{s=1}^{t} \alpha_s$. It follows that the marginal distribution admits the closed form:

$$q(\mathbf{x}_t \mid \mathbf{x}_0) = \mathcal{N}\left(\mathbf{x}_t; \sqrt{\bar{\alpha}_t}\,\mathbf{x}_0, (1 - \bar{\alpha}_t)\,\mathbf{I}\right), \tag{25}$$

and equivalently that

$$\mathbf{x}_t = \sqrt{\bar{\alpha}_t}\,\mathbf{x}_0 + \sigma_t\,\epsilon, \; \sigma_t^2 := 1 - \bar{\alpha}_t. \tag{26}$$

## A.2. Denoising-score identity: $\mathbb{E}[\epsilon \mid \mathbf{x}_t]$ and applying the forward process in $\mathbf{s}_t(\mathbf{x}_t)$

Let $p_t$ denote the marginal density of $\mathbf{x}_t$ induced by $\mathbf{x}_0 \sim p_0$ and (26). We define the ideal score as $\mathbf{s}_t(\mathbf{x}) := \nabla_{\mathbf{x}} \log p_t(\mathbf{x})$ (Vincent, 2011; Saremi & Hyvärinen, 2019).

**Lemma A.1** (Conditional-noise / score identity). *For each fixed $t$,*

$$\mathbf{s}_t(\mathbf{x}_t) = -\frac{1}{\sigma_t}\,\mathbb{E}[\epsilon \mid \mathbf{x}_t]. \tag{27}$$

*Proof.* Let $K_t(\mathbf{x}_t \mid \mathbf{x}_0) = \mathcal{N}\left(\mathbf{x}_t; \sqrt{\bar{\alpha}_t}\,\mathbf{x}_0, \sigma_t^2\,\mathbf{I}\right)$ denote the Gaussian transition kernel of the forward process. The marginal density of $\mathbf{x}_t$ can then be written as

$$p_t(\mathbf{x}_t) = \int p_0(\mathbf{x}_0)\,K_t(\mathbf{x}_t \mid \mathbf{x}_0)\,d\mathbf{x}_0.$$

Differentiating under the integral yields:

$$\nabla_{\mathbf{x}_t} p_t(\mathbf{x}_t) = \int p_0(\mathbf{x}_0)\,K_t(\mathbf{x}_t \mid \mathbf{x}_0)\,\nabla_{\mathbf{x}_t} \log K_t(\mathbf{x}_t \mid \mathbf{x}_0)\,d\mathbf{x}_0.$$

Since

$$\nabla_{\mathbf{x}_t} \log K_t(\mathbf{x}_t \mid \mathbf{x}_0) = -\frac{\mathbf{x}_t - \sqrt{\bar{\alpha}_t}\,\mathbf{x}_0}{\sigma_t^2}.$$

We obtain:

$$\nabla_{\mathbf{x}_t} \log p_t(\mathbf{x}_t) = \frac{\nabla p_t(\mathbf{x}_t)}{p_t(\mathbf{x}_t)} = -\mathbb{E}\left[\frac{\mathbf{x}_t - \sqrt{\bar{\alpha}_t}\,\mathbf{x}_0}{\sigma_t^2}\,\Big|\,\mathbf{x}_t\right].$$

Using the identity, $\epsilon = (\mathbf{x}_t - \sqrt{\bar{\alpha}_t}\,\mathbf{x}_0)/\sigma_t$, this simplifies to

$$\nabla_{\mathbf{x}_t} \log p_t(\mathbf{x}_t) = -\frac{1}{\sigma_t}\mathbb{E}[\epsilon \mid \mathbf{x}_t],$$

which establishes (27). $\square$

**Implication for $\epsilon$-prediction.** By definition of the mean squared error objective, $\epsilon_\theta(\mathbf{x}_t, t)$ is an estimator of $\mathbb{E}[\epsilon \mid \mathbf{x}_t]$. Combining this observation with (27) yields

$$\mathbf{s}_\theta(\mathbf{x}_t, t) = -\frac{1}{\sigma_t}\,\epsilon_\theta(\mathbf{x}_t, t). \tag{28}$$

### A.3. Tweedie formula under DDPM scaling (posterior mean of $\mathbf{x}_0$)

The classical Tweedie formula is for additive noise model $\mathbf{y} = \mathbf{x} + \sigma\,\boldsymbol{\epsilon}$ (Robbins, 1956; Efron, 2011). whereas DDPM involves an additional scaling factor $\sqrt{\bar{\alpha}_t}$. We therefore reduce to the additive setting by introducing a rescaled variable:

$$\mathbf{y}_t := \frac{\mathbf{x}_t}{\sqrt{\bar{\alpha}_t}} = \mathbf{x}_0 + \tilde{\sigma}_t\,\boldsymbol{\epsilon}, \qquad \tilde{\sigma}_t := \frac{\sigma_t}{\sqrt{\bar{\alpha}_t}}. \tag{29}$$

Let $\tilde{p}_t$ denote the marginal density of $\mathbf{y}_t$ and define its score by $\tilde{\mathbf{s}}_t(\mathbf{y}) := \nabla_{\mathbf{y}} \log \tilde{p}_t(\mathbf{y})$. For clarity, in the next two subsections we work with a generic additive Gaussian model $\mathbf{y} = \mathbf{x}_0 + \tilde{\sigma}\,\boldsymbol{\epsilon}$ and omit the time index $t$, writing $\tilde{p}$ and $\tilde{\mathbf{s}}$ for the corresponding marginal and score.

**Lemma A.2** (Tweedie (additive form)). *For the additive noise model* $\mathbf{y} = \mathbf{x}_0 + \tilde{\sigma}\,\boldsymbol{\epsilon}$, *we have*

$$\mathbb{E}[\mathbf{x}_0 \mid \mathbf{y}] = \mathbf{y} + \tilde{\sigma}^2\,\tilde{\mathbf{s}}(\mathbf{y}). \tag{30}$$

*Proof.* Applying Lemma A.1 in the additive model gives $\tilde{\mathbf{s}}(\mathbf{y}) = -(1/\tilde{\sigma})\,\mathbb{E}[\boldsymbol{\epsilon} \mid \mathbf{y}]$ and $\mathbf{x}_0 = \mathbf{y} - \tilde{\sigma}\,\boldsymbol{\epsilon}$. Taking the conditional expectation yields $\mathbb{E}[\mathbf{x}_0 \mid \mathbf{y}] = \mathbf{y} - \tilde{\sigma}\,\mathbb{E}[\boldsymbol{\epsilon} \mid \mathbf{y}] = \mathbf{y} + \tilde{\sigma}^2\,\tilde{\mathbf{s}}(\mathbf{y})$, which establishes (30). $\square$

We now translate the Tweedie formula back to $\mathbf{x}_t$. Since $\mathbf{y}_t = \mathbf{x}_t/\sqrt{\bar{\alpha}_t}$, the score transforms by the chain rule:

$$\tilde{\mathbf{s}}_t(\mathbf{y}_t) = \nabla_{\mathbf{y}_t} \log \tilde{p}_t(\mathbf{y}_t) = \sqrt{\bar{\alpha}_t}\,\nabla_{\mathbf{x}_t} \log p_t(\mathbf{x}_t) = \sqrt{\bar{\alpha}_t}\,\mathbf{s}_t(\mathbf{x}_t). \tag{31}$$

Combining (30) and (31) gives the DDPM-scaled Tweedie formula:

**Lemma A.3** (Tweedie for DDPM). *Let* $m(\mathbf{x}_t) := \mathbb{E}[\mathbf{x}_0 \mid \mathbf{x}_t]$ *denote the Bayes denoiser (posterior mean). Then*

$$m(\mathbf{x}_t) = \frac{1}{\sqrt{\bar{\alpha}_t}}\Big(\mathbf{x}_t + \sigma_t^2\,\mathbf{s}_t(\mathbf{x}_t)\Big). \tag{32}$$

### A.4. Posterior covariance and Jacobian: $\mathrm{Cov}(\mathbf{x}_0 \mid \mathbf{x}_t)$

This subsection makes explicit the identity "Jacobian = posterior covariance" that underlies Lipschitz and contractivity arguments (Saremi & Hyvärinen, 2019; Guo et al., 2005). We work in the additive form $\mathbf{y} = \mathbf{x}_0 + \tilde{\sigma}\,\boldsymbol{\epsilon}$ for clarity. Let $m(\mathbf{y}) := \mathbb{E}[\mathbf{x}_0 \mid \mathbf{y}]$ and $C(\mathbf{y}) := \mathrm{Cov}(\mathbf{x}_0 \mid \mathbf{y})$.

**Lemma A.4** (Posterior covariance identity). *For additive Gaussian noise,*

$$C(\mathbf{y}) = \tilde{\sigma}^2\,\mathbf{I} + \tilde{\sigma}^4\,\nabla_{\mathbf{y}}^2 \log \tilde{p}(\mathbf{y}), \tag{33}$$

*and equivalently, using* $m(\mathbf{y}) = \mathbf{y} + \tilde{\sigma}^2\,\nabla_{\mathbf{y}} \log \tilde{p}(\mathbf{y})$,

$$\nabla_{\mathbf{y}} m(\mathbf{y}) = \mathbf{I} + \tilde{\sigma}^2\,\nabla_{\mathbf{y}}^2 \log \tilde{p}(\mathbf{y}), \qquad C(\mathbf{y}) = \tilde{\sigma}^2\,\nabla_{\mathbf{y}} m(\mathbf{y}). \tag{34}$$

*Proof.* We start from the posterior mean expressed as

$$m(\mathbf{y}) = \frac{1}{\tilde{p}(\mathbf{y})} \int \mathbf{x}\,p_0(\mathbf{x})\,\phi_{\tilde{\sigma}}(\mathbf{y} - \mathbf{x})\,d\mathbf{x}\,,$$

where $\phi_{\tilde{\sigma}}(.)$ is the Gaussian density with variance $\tilde{\sigma}^2\,\mathbf{I}$. Differentiating componentwise with respect to $\mathbf{y}$ and using $\nabla_{\mathbf{y}} \phi_{\tilde{\sigma}}(\mathbf{y} - \mathbf{x}) = -(\mathbf{y} - \mathbf{x})\phi_{\tilde{\sigma}}(\mathbf{y} - \mathbf{x})/\tilde{\sigma}^2$, a standard quotient-rule calculation gives

$$\nabla_{\mathbf{y}} m(\mathbf{y}) = \frac{1}{\tilde{\sigma}^2}\Big(\mathbb{E}[\mathbf{x}_0 \mathbf{x}_0^\top \mid \mathbf{y}] - \mathbb{E}[\mathbf{x}_0 \mid \mathbf{y}]\,\mathbb{E}[\mathbf{x}_0 \mid \mathbf{y}]^\top\Big) = \frac{1}{\tilde{\sigma}^2}\,C(\mathbf{y})\,.$$

This immediately yields $C(\mathbf{y}) = \tilde{\sigma}^2\,\nabla_{\mathbf{y}} m(\mathbf{y})$.

To obtain (33), differentiate the Tweedie formula $m(\mathbf{y}) = \mathbf{y} + \tilde{\sigma}^2\,\nabla_{\mathbf{y}} \log \tilde{p}(\mathbf{y})$ to get $\nabla_{\mathbf{y}} m(\mathbf{y}) = \mathbf{I} + \tilde{\sigma}^2\,\nabla_{\mathbf{y}}^2 \log \tilde{p}(\mathbf{y})$ and multiply both sides by $\tilde{\sigma}^2$. $\square$

**DDPM scaling.** For the forward sample $\mathbf{x}_t = \sqrt{\bar{\alpha}_t}\,\mathbf{x}_0 + \sigma_t\,\boldsymbol{\epsilon}$, define the rescaled variable $\mathbf{y}_t = \mathbf{x}_t/\sqrt{\bar{\alpha}_t}$. Then $\tilde{\sigma}_t = \sigma_t/\sqrt{\bar{\alpha}_t}$ and the same identities hold for the posterior of $\mathbf{x}_0 \mid \mathbf{x}_t$ after change of variables.

## A.5. Lipschitzness and contractivity of the Bayes denoiser

The identity $C(\mathbf{y}) = \tilde{\sigma}^2\,\nabla_{\mathbf{y}} m(\mathbf{y})$ immediately provides Lipschitz control of the posterior mean. Such covariance bounds hold, for example, under (strong) log-concavity of the prior via Brascamp–Lieb inequalities (Brascamp & Lieb, 1976).

**Lemma A.5** (Covariance bound implies Lipschitz denoiser). *Let $\Omega \subset \mathbb{R}^d$ be a region where the posterior covariance satisfies $\|C(\mathbf{y})\|_{\mathrm{op}} \leq \rho\,\tilde{\sigma}^2$ for all $\mathbf{y} \in \Omega$, then the posterior mean $m(.)$ satisfies $\|\nabla_{\mathbf{y}} m(\mathbf{y})\|_{\mathrm{op}} \leq \rho$ for all $\mathbf{y} \in \Omega$ and is therefore $\rho$-Lipschitz on $\Omega$.*

*Proof.* Using the identity $C(\mathbf{y}) = \tilde{\sigma}^2\,\nabla_{\mathbf{y}} m(\mathbf{y})$, and taking operator norms, we have $\|\nabla_{\mathbf{y}} m(\mathbf{y})\|_{\mathrm{op}} = \|C(\mathbf{y})\|_{\mathrm{op}}/\tilde{\sigma}^2 \leq \rho$, which establishes the Lipschitz bound. $\qquad\square$

For directional contraction—used in the cross-backbone "normal-to-manifold" argument—we isolate a normal direction $\mathbf{n}$ and assume contraction along that direction.

**Assumption A.6** (Directional contraction of the denoiser). (Dalalyan, 2017; Durmus & Moulines, 2019) There exists $\kappa \in (0, 1]$ such that, for all $\mathbf{y}, \mathbf{y}'$ in the tube,

$$\langle m(\mathbf{y}) - m(\mathbf{y}'),\, \mathbf{y} - \mathbf{y}' \rangle \leq (1 - \kappa)\,\|\mathbf{y} - \mathbf{y}'\|_2^2 \qquad \text{whenever } (\mathbf{y} - \mathbf{y}') \parallel \mathbf{n}. \tag{35}$$

A sufficient condition is (locally, a.e.) a bound on the directional derivative along $\mathbf{n}$ in the tube: $\langle \mathbf{n}, (\nabla_{\mathbf{y}} m(\mathbf{y}))\,\mathbf{n} \rangle \leq 1 - \kappa$. Using the covariance–Jacobian identity (34), this is equivalent to $\langle \mathbf{n}, C(\mathbf{y})\mathbf{n} \rangle \leq (1 - \kappa)\,\tilde{\sigma}^2$.

## A.6. From denoiser contraction to directional growth of the score

This is the key step used to justify the main-text condition (18). We work in additive coordinates, $\mathbf{y} = \mathbf{x}_0 + \tilde{\sigma}\,\boldsymbol{\epsilon}$. From the Tweedie formula, $\tilde{\mathbf{s}}(\mathbf{y}) \coloneqq \nabla_{\mathbf{y}} \log \tilde{p}(\mathbf{y})$, Let $\mathbf{y}'$ denote a projection point (Alain & Bengio, 2014) (e.g., $\mathbf{y}' = \pi(\mathbf{y})$) and define $\mathbf{v} = \mathbf{y} - \mathbf{y}'$.

**Lemma A.7** (Directional growth of the ideal score). *Assume (35) holds for $\mathbf{y}, \mathbf{y}'$ with $\mathbf{v} \parallel \mathbf{n}$. Then*

$$\left\langle \tilde{\mathbf{s}}(\mathbf{y}) - \tilde{\mathbf{s}}(\mathbf{y}'),\, \frac{\mathbf{v}}{\|\mathbf{v}\|} \right\rangle \leq -\frac{\kappa}{\tilde{\sigma}^2}\,\|\mathbf{v}\|. \tag{36}$$

*Proof.* Using $\tilde{\mathbf{s}}(\mathbf{y}) = (m(\mathbf{y}) - \mathbf{y})/\tilde{\sigma}^2$, we have

$$\tilde{\mathbf{s}}(\mathbf{y}) - \tilde{\mathbf{s}}(\mathbf{y}') = \frac{(m(\mathbf{y}) - m(\mathbf{y}')) - (\mathbf{y} - \mathbf{y}')}{\tilde{\sigma}^2} \,.$$

Taking the inner product with $\mathbf{v} = \mathbf{y} - \mathbf{y}'$ gives:

$$\langle \tilde{\mathbf{s}}(\mathbf{y}) - \tilde{\mathbf{s}}(\mathbf{y}'),\, \mathbf{v} \rangle = \frac{\langle m(\mathbf{y}) - m(\mathbf{y}'), \mathbf{v} \rangle - \|\mathbf{v}\|^2}{\tilde{\sigma}^2} \leq -\frac{\kappa}{\tilde{\sigma}^2}\,\|\mathbf{v}\|^2 \,,$$

by Assumption A.6. Dividing both sides by $\|\mathbf{v}\|$ yields (36). $\qquad\square$

## A.7. From $s$ to $\mathbf{s}_\theta$ (approximation on a tube)

Let $\mathbf{s}_\theta$ be a learned score that approximates the source score on a tube:

$$\sup_{\mathbf{y} \in \Omega} \|\mathbf{s}_\theta(\mathbf{y}) - \tilde{\mathbf{s}}(\mathbf{y})\| \leq \delta \,.$$

Then the directional inequality transfers with a slack.

**Lemma A.8** (Directional growth for $\mathbf{s}_\theta$). *Under the above uniform approximation, for $\mathbf{v} = \mathbf{y} - \mathbf{y}'$,*

$$\left\langle \mathbf{s}_\theta(\mathbf{y}) - \mathbf{s}_\theta(\mathbf{y}'), \frac{\mathbf{v}}{\|\mathbf{v}\|} \right\rangle \le -\frac{\kappa}{\tilde{\sigma}^2}\|\mathbf{v}\| + 2\delta. \tag{37}$$

*In particular, if $\|\mathbf{v}\| \ge \dfrac{4\tilde{\sigma}^2}{\kappa}\delta$, then*

$$\left\langle \mathbf{s}_\theta(\mathbf{y}) - \mathbf{s}_\theta(\mathbf{y}'), \frac{\mathbf{v}}{\|\mathbf{v}\|} \right\rangle \le -\underline{m}\|\mathbf{v}\|, \qquad \underline{m} := \frac{\kappa}{2\tilde{\sigma}^2}. \tag{38}$$

*Proof.* Decompose $\mathbf{s}_\theta(\mathbf{y}) = \tilde{\mathbf{s}}(\mathbf{y}) + \boldsymbol{\xi}(\mathbf{y})$ with $\|\boldsymbol{\xi}(\mathbf{y})\| \le \delta$:

$$\langle \mathbf{s}_\theta(\mathbf{y}) - \mathbf{s}_\theta(\mathbf{y}'), \mathbf{v}/\|\mathbf{v}\| \rangle = \langle \tilde{\mathbf{s}}(\mathbf{y}) - \tilde{\mathbf{s}}(\mathbf{y}'), \mathbf{v}/\|\mathbf{v}\| \rangle + \langle \boldsymbol{\xi}(\mathbf{y}) - \boldsymbol{\xi}(\mathbf{y}'), \mathbf{v}/\|\mathbf{v}\| \rangle.$$

By Lemma A.7 and $|\langle \boldsymbol{\xi}(\mathbf{y}) - \boldsymbol{\xi}(\mathbf{y}'), \cdot \rangle| \le \|\boldsymbol{\xi}(\mathbf{y})\| + \|\boldsymbol{\xi}(\mathbf{y}')\| \le 2\delta$, we get (37). If $\|\mathbf{v}\| \ge 4\tilde{\sigma}^2\delta/\kappa$ then $2\delta \le (\kappa/(2\tilde{\sigma}^2))\|\mathbf{v}\|$ and (38) follows. $\square$

**Connection to the main-text condition** (18). In the main text, a projection $\pi_t$ onto a source manifold $\mathcal{M}_t$ is defined in $\mathbf{x}_t$-space. Applying the previous derivation in the rescaled additive coordinates $\mathbf{y}_t = \mathbf{x}_t/\sqrt{\bar{\alpha}_t}$ yields (18) with explicit definitions of $m_t$ and $d_{0,t}$ up to the scaling $\tilde{\sigma}_t = \sigma_t/\sqrt{\bar{\alpha}_t}$.

# B. GEPC theory: detailed proofs and cross-backbone geometry

## B.1. Invariance of a distribution and score equivariance

We work with a finite group $\mathcal{G}$ acting on $\mathbb{R}^d$ via orthogonal matrices $\mathcal{P}_g$, so $\mathcal{P}_g^{-1} = \mathcal{P}_g^\top$ and $|\det \mathcal{P}_g| = 1$.

**Lemma B.1** (Invariance $\Leftrightarrow$ score equivariance). *Let $p$ be a positive $C^1$ density on $\mathbb{R}^d$ with score $\mathbf{s}_p(\mathbf{x}) = \nabla_\mathbf{x} \log p(\mathbf{x})$. Then the following are equivalent:*

*(i) $p(\mathcal{P}_g\mathbf{x}) = p(\mathbf{x})$ for all $g \in \mathcal{G}$ and all $\mathbf{x} \in \mathbb{R}^d$;*

*(ii) $\mathbf{s}_p(\mathcal{P}_g\mathbf{x}) = \mathcal{P}_g\mathbf{s}_p(\mathbf{x})$ for all $g \in \mathcal{G}$ and all $\mathbf{x} \in \mathbb{R}^d$.*

*Proof.* (i)$\Rightarrow$(ii). If $p(\mathcal{P}_g\mathbf{x}) = p(\mathbf{x})$, then $\log p(\mathcal{P}_g\mathbf{x}) = \log p(\mathbf{x})$. Differentiating w.r.t. $\mathbf{x}$ and using the chain rule gives

$$\nabla_\mathbf{x} \log p(\mathcal{P}_g\mathbf{x}) = \mathcal{P}_g^\top \nabla_\mathbf{y} \log p(\mathbf{y})\big|_{\mathbf{y} = \mathcal{P}_g\mathbf{x}} = \mathcal{P}_g^\top \mathbf{s}_p(\mathcal{P}_g\mathbf{x}).$$

The left-hand side equals $\nabla_\mathbf{x} \log p(\mathbf{x}) = \mathbf{s}_p(\mathbf{x})$, hence $\mathbf{s}_p(\mathbf{x}) = \mathcal{P}_g^\top \mathbf{s}_p(\mathcal{P}_g\mathbf{x})$, i.e. $\mathbf{s}_p(\mathcal{P}_g\mathbf{x}) = \mathcal{P}_g\mathbf{s}_p(\mathbf{x})$.

(ii)$\Rightarrow$(i). Assume $\mathbf{s}_p(\mathcal{P}_g\mathbf{x}) = \mathcal{P}_g\mathbf{s}_p(\mathbf{x})$. Define $h_g(\mathbf{x}) := \log p(\mathcal{P}_g\mathbf{x}) - \log p(\mathbf{x})$. Then

$$\nabla_\mathbf{x} h_g(\mathbf{x}) = \mathcal{P}_g^\top \mathbf{s}_p(\mathcal{P}_g\mathbf{x}) - \mathbf{s}_p(\mathbf{x}) = \mathcal{P}_g^\top \mathcal{P}_g \mathbf{s}_p(\mathbf{x}) - \mathbf{s}_p(\mathbf{x}) = \mathbf{0},$$

so $h_g(\mathbf{x})$ is constant in $\mathbf{x}$: $h_g(\mathbf{x}) = c_g$. Hence $p(\mathcal{P}_g\mathbf{x}) = e^{c_g}p(\mathbf{x})$. Integrating both sides over $\mathbb{R}^d$ and using $|\det \mathcal{P}_g| = 1$ yields $1 = \int p(\mathcal{P}_g\mathbf{x})d\mathbf{x} = e^{c_g}\int p(\mathbf{x})d\mathbf{x} = e^{c_g}$, so $c_g = 0$ and $p(\mathcal{P}_g\mathbf{x}) = p(\mathbf{x})$. $\square$

## B.2. Residual decomposition and expectation bounds

Recall the residual operator from (9):

$$\Delta_g f(\mathbf{x}, t) := \mathcal{P}_g^{-1} f(\mathcal{P}_g\mathbf{x}, t) - f(\mathbf{x}, t).$$

For orthogonal transforms, $\mathcal{P}_g^{-1} = \mathcal{P}_g^\top$ and $\|\mathcal{P}_g^{-1}\mathbf{v}\|_2 = \|\mathbf{v}\|_2$. For a backbone score $\mathbf{s}_\theta(\cdot, t)$ we define

$$R_t(\mathbf{x}, g) := \|\Delta_g \mathbf{s}_\theta(\mathbf{x}, t)\|_2^2. \tag{39}$$

Fix any absolutely continuous test marginal $p_t$ with score $\mathbf{s}_{p_t}(\mathbf{x}) := \nabla_{\mathbf{x}} \log p_t(\mathbf{x})$ and define the score error $\mathbf{e}_{p_t}(\mathbf{x}, t) := \mathbf{s}_\theta(\mathbf{x}, t) - \mathbf{s}_{p_t}(\mathbf{x})$. Then for all $\mathbf{x}, g$,

$$\Delta_g \mathbf{s}_\theta(\mathbf{x}, t) = \Delta_g \mathbf{s}_{p_t}(\mathbf{x}, t) + \Delta_g \mathbf{e}_{p_t}(\mathbf{x}, t), \qquad R_t(\mathbf{x}, g) = \|\Delta_g \mathbf{s}_{p_t}(\mathbf{x}, t) + \Delta_g \mathbf{e}_{p_t}(\mathbf{x}, t)\|_2^2. \tag{40}$$

We also recall

$$\mathcal{B}^{(\mathcal{G})}(p_t) := \mathbb{E}_{\mathbf{x} \sim p_t, \, g \sim \nu_\mathcal{G}} \big[ \|\Delta_g \mathbf{s}_{p_t}(\mathbf{x}, t)\|_2^2 \big], \tag{41}$$

and define

$$\Delta_E(p_t, t) := \mathbb{E}_{\mathbf{x} \sim p_t, \, g \sim \nu_\mathcal{G}} \Big[ \|\mathbf{e}_{p_t}(\mathcal{P}_g \mathbf{x}, t) - \mathbf{e}_{p_t}(\mathbf{x}, t)\|_2^2 \Big]. \tag{42}$$

**Proof of Proposition 4.2.** Expanding the squared norm in (40) gives

$$R_t(\mathbf{x}, g) = \|\Delta_g \mathbf{s}_{p_t}(\mathbf{x}, t)\|_2^2 + \|\Delta_g \mathbf{e}_{p_t}(\mathbf{x}, t)\|_2^2 + 2\langle \Delta_g \mathbf{s}_{p_t}(\mathbf{x}, t), \, \Delta_g \mathbf{e}_{p_t}(\mathbf{x}, t) \rangle. \tag{43}$$

*Upper bound.* Using Cauchy–Schwarz and inequality $2\langle \mathbf{a}, \mathbf{b} \rangle \le \|\mathbf{a}\|_2^2 + \|\mathbf{b}\|_2^2$, for any vectors $\mathbf{a}$ and $\mathbf{b}$, in (43) leads to

$$R_t(\mathbf{x}, g) \le 2\|\Delta_g \mathbf{s}_{p_t}(\mathbf{x}, t)\|_2^2 + 2\|\Delta_g \mathbf{e}_{p_t}(\mathbf{x}, t)\|_2^2.$$

Taking expectation over $\mathbf{x} \sim p_t$ and $g \sim \nu_\mathcal{G}$ yields

$$\mathbb{E}[R_t(\mathbf{x}, g)] \le 2\mathcal{B}^{(\mathcal{G})}(p_t) + 2\,\mathbb{E}\|\Delta_g \mathbf{e}_{p_t}(\mathbf{x}, t)\|_2^2.$$

Finally, since $\Delta_g \mathbf{e}_{p_t}(\mathbf{x}, t) = \mathcal{P}_g^{-1} \mathbf{e}_{p_t}(\mathcal{P}_g \mathbf{x}, t) - \mathbf{e}_{p_t}(\mathbf{x}, t)$ and $\|\mathbf{u} - \mathbf{v}\|_2^2 \le 2\|\mathbf{u}\|_2^2 + 2\|\mathbf{v}\|_2^2$,

$$\|\Delta_g \mathbf{e}_{p_t}(\mathbf{x}, t)\|_2^2 \le 2\|\mathbf{e}_{p_t}(\mathcal{P}_g \mathbf{x}, t)\|_2^2 + 2\|\mathbf{e}_{p_t}(\mathbf{x}, t)\|_2^2,$$

which gives the stated $u_b(p_t)$ in Proposition 4.2.

*Lower bound.* From (43) and Cauchy–Schwarz,

$$R_t(\mathbf{x}, g) \ge \|\Delta_g \mathbf{s}_{p_t}(\mathbf{x}, t)\|_2^2 + \|\Delta_g \mathbf{e}_{p_t}(\mathbf{x}, t)\|_2^2 - 2\|\Delta_g \mathbf{s}_{p_t}(\mathbf{x}, t)\|_2 \, \|\Delta_g \mathbf{e}_{p_t}(\mathbf{x}, t)\|_2.$$

Taking expectation and applying Cauchy–Schwarz to the cross term yields

$$\mathbb{E}[R_t(\mathbf{x}, g)] \ge \mathcal{B}^{(\mathcal{G})}(p_t) + \mathbb{E}\|\Delta_g \mathbf{e}_{p_t}(\mathbf{x}, t)\|_2^2 - 2\sqrt{\mathcal{B}^{(\mathcal{G})}(p_t)} \sqrt{\mathbb{E}\|\Delta_g \mathbf{e}_{p_t}(\mathbf{x}, t)\|_2^2}.$$

Noting that $\Delta_g \mathbf{e}_{p_t}(\mathbf{x}, t) = \mathcal{P}_g^{-1} \mathbf{e}_{p_t}(\mathcal{P}_g \mathbf{x}, t) - \mathbf{e}_{p_t}(\mathbf{x}, t)$ and $\|\mathcal{P}_g^{-1} \mathbf{v}\|_2 = \|\mathbf{v}\|_2$, we obtain $\mathbb{E}\|\Delta_g \mathbf{e}_{p_t}(\mathbf{x}, t)\|_2^2 = \Delta_E(p_t, t)$, which yields the lower bound of proposition 4.2. $\square$

### B.3. Cross-backbone bounds: proof of Proposition 4.3

Fix $t$ and consider $\mathbf{x} \in \mathcal{N}_t$. Let $\mathbf{z} = \pi_t(\mathbf{x}) \in \mathcal{M}_t$, so that $d_t(\mathbf{x}) = \|\mathbf{x} - \mathbf{z}\|_2$. Assume $\pi_t$ commutes with the group action: $\pi_t(\mathcal{P}_g \mathbf{x}) = \mathcal{P}_g \mathbf{z}$, and $\mathcal{P}_g$ is orthogonal, so $\|\mathcal{P}_g \mathbf{x} - \mathcal{P}_g \mathbf{z}\|_2 = \|\mathbf{x} - \mathbf{z}\|_2 = d_t(\mathbf{x})$.

Define the off-manifold deviation $\boldsymbol{\delta}(\mathbf{x}) := \mathbf{s}_\theta(\mathbf{x}, t) - \mathbf{s}_\theta(\mathbf{z}, t)$. By Lipschitzness (16), $\|\boldsymbol{\delta}(\mathbf{x})\|_2 \le L_t d_t(\mathbf{x})$ and $\|\boldsymbol{\delta}(\mathcal{P}_g \mathbf{x})\|_2 \le L_t d_t(\mathbf{x})$.

**Upper bound (17).** Using add-and-subtract around $\mathbf{z}$ and $\mathcal{P}_g \mathbf{z}$:

$$\begin{aligned}
\Delta_g \mathbf{s}_\theta(\mathbf{x}, t) &= \mathcal{P}_g^\top \mathbf{s}_\theta(\mathcal{P}_g \mathbf{x}, t) - \mathbf{s}_\theta(\mathbf{x}, t) \\
&= \underbrace{\big( \mathcal{P}_g^\top \mathbf{s}_\theta(\mathcal{P}_g \mathbf{z}, t) - \mathbf{s}_\theta(\mathbf{z}, t) \big)}_{\Delta_g \mathbf{s}_\theta(\mathbf{z}, t)} + \underbrace{\mathcal{P}_g^\top \big( \mathbf{s}_\theta(\mathcal{P}_g \mathbf{x}, t) - \mathbf{s}_\theta(\mathcal{P}_g \mathbf{z}, t) \big) - \big( \mathbf{s}_\theta(\mathbf{x}, t) - \mathbf{s}_\theta(\mathbf{z}, t) \big)}_{\mathbf{b}_g(\mathbf{x})}.
\end{aligned}$$

Thus, $R_t(\mathbf{x}, g) = \|\Delta_g \mathbf{s}_\theta(\mathbf{z}, t) + \mathbf{b}_g(\mathbf{x})\|_2^2$. Using the inequality $\|\mathbf{a} + \mathbf{b}\|_2^2 \le 2\|\mathbf{a}\|_2^2 + 2\|\mathbf{b}\|_2^2$ for any vectors $\mathbf{a}$ and $\mathbf{b}$ gives

$$R_t(\mathbf{x}, g) \le 2R_t(\mathbf{z}, g) + 2\|\mathbf{b}_g(\mathbf{x})\|_2^2.$$

Moreover, by the triangle inequality and orthogonality of $\mathcal{P}_g$,

$$\|\mathbf{b}_g(\mathbf{x})\|_2 \leq \|\mathbf{s}_\theta(\mathcal{P}_g\mathbf{x}, t) - \mathbf{s}_\theta(\mathcal{P}_g\mathbf{z}, t)\|_2 + \|\mathbf{s}_\theta(\mathbf{x}, t) - \mathbf{s}_\theta(\mathbf{z}, t)\|_2 \leq 2L_t d_t(\mathbf{x}),$$

so $\|\mathbf{b}_g(\mathbf{x})\|_2^2 \leq 4L_t^2 d_t(\mathbf{x})^2$ and therefore

$$R_t(\mathbf{x}, g) \leq 2R_t(\mathbf{z}, g) + 8L_t^2 d_t(\mathbf{x})^2.$$

Taking expectation over $g \sim \nu_{\mathcal{G}}$ yields (17).

**Lower bound** (19). Let $\mathbf{z} = \pi_t(\mathbf{x})$. By the reverse triangle inequality,

$$\|\Delta_g\mathbf{s}_\theta(\mathbf{x}, t)\|_2 = \|\mathcal{P}_g^\top\mathbf{s}_\theta(\mathcal{P}_g\mathbf{x}, t) - \mathbf{s}_\theta(\mathbf{x}, t)\|_2 \geq \|\mathbf{s}_\theta(\mathbf{x}, t) - \mathbf{s}_\theta(\mathbf{z}, t)\|_2 - \|\mathcal{P}_g^\top\mathbf{s}_\theta(\mathcal{P}_g\mathbf{x}, t) - \mathbf{s}_\theta(\mathbf{z}, t)\|_2.$$

Using $\|\mathcal{P}_g^\top\mathbf{u} - \mathbf{v}\|_2 = \|\mathbf{u} - \mathcal{P}_g\mathbf{v}\|_2$ and adding/subtracting $\mathbf{s}_\theta(\mathcal{P}_g\mathbf{z}, t)$,

$$\|\mathcal{P}_g^\top\mathbf{s}_\theta(\mathcal{P}_g\mathbf{x}, t) - \mathbf{s}_\theta(\mathbf{z}, t)\|_2 = \|\mathbf{s}_\theta(\mathcal{P}_g\mathbf{x}, t) - \mathcal{P}_g\mathbf{s}_\theta(\mathbf{z}, t)\|_2 \leq \|\mathbf{s}_\theta(\mathcal{P}_g\mathbf{x}, t) - \mathbf{s}_\theta(\mathcal{P}_g\mathbf{z}, t)\|_2 + \|\Delta_g\mathbf{s}_\theta(\mathbf{z}, t)\|_2 \leq L_t d_t(\mathbf{x}) + \|\Delta_g\mathbf{s}_\theta(\mathbf{z}, t)\|_2.$$

Hence,

$$\|\Delta_g\mathbf{s}_\theta(\mathbf{x}, t)\|_2 \geq \|\mathbf{s}_\theta(\mathbf{x}, t) - \mathbf{s}_\theta(\mathbf{z}, t)\|_2 - L_t d_t(\mathbf{x}) - \|\Delta_g\mathbf{s}_\theta(\mathbf{z}, t)\|_2.$$

If $d_t(\mathbf{x}) \geq d_{0,t}$ and (18) holds, then $\|\mathbf{s}_\theta(\mathbf{x}, t) - \mathbf{s}_\theta(\mathbf{z}, t)\|_2 \geq m_t d_t(\mathbf{x})$ (since $\|v\| \geq |\langle v, u\rangle|$ for unit $u$). Thus,

$$\|\Delta_g\mathbf{s}_\theta(\mathbf{x}, t)\|_2 \geq (m_t - L_t)\,d_t(\mathbf{x}) - \|\Delta_g\mathbf{s}_\theta(\mathbf{z}, t)\|_2.$$

Let $a := (m_t - L_t)\,d_t(\mathbf{x})$ and $\mathbf{z} := \pi_t(\mathbf{x})$. From the previous pointwise inequality, for each $g \in \mathcal{G}$,

$$R_t(\mathbf{x}, g) \geq (a - \|\Delta_g\mathbf{s}_\theta(\mathbf{z}, t)\|_2)_+^2.$$

Define $\varphi(y) := (a - y)_+^2$, which is convex. By Jensen's inequality (Boyd & Vandenberghe, 2004),

$$\mathbb{E}_g R_t(\mathbf{x}, g) \geq \mathbb{E}_g \varphi(\|\Delta_g\mathbf{s}_\theta(\mathbf{z}, t)\|_2) \geq \varphi(\mathbb{E}_g \|\Delta_g\mathbf{s}_\theta(\mathbf{z}, t)\|_2).$$

Moreover, by Cauchy–Schwarz,

$$\mathbb{E}_g \|\Delta_g\mathbf{s}_\theta(\mathbf{z}, t)\|_2 \leq \sqrt{\mathbb{E}_g \|\Delta_g\mathbf{s}_\theta(\mathbf{z}, t)\|_2^2} = \sqrt{\mathbb{E}_g R_t(\mathbf{z}, g)} = \rho_t(\mathbf{x}).$$

Since $\varphi$ is non-increasing on $\mathbb{R}_+$, we obtain

$$\mathbb{E}_g R_t(\mathbf{x}, g) \geq \varphi(\rho_t(\mathbf{x})) = \left(((m_t - L_t)d_t(\mathbf{x}) - \rho_t(\mathbf{x}))_+\right)^2,$$

where $(a)_+ := \max\{a, 0\}$. This proves the claimed lower bound in Proposition 4.3. $\qquad\square$

## C. Gaussian sanity checks (mean shift and $90°$ rotation)

We provide closed-form computations of the *ideal* GEPC residual for a simple Gaussian, illustrating that GEPC captures equivariance-breaking information even when the score magnitude remains insensitive.

**C.1. Mean shift with $\mathcal{G} = \{\mathbf{I}_d, -\mathbf{I}_d\}$**

Let $p = \mathcal{N}(\boldsymbol{\mu}, \sigma^2\mathbf{I})$. Then $s(\mathbf{x}) = \nabla_{\mathbf{x}} \log p(\mathbf{x}) = -(\mathbf{x} - \boldsymbol{\mu})/\sigma^2$. For $\mathcal{G} = \{\mathbf{I}_d, -\mathbf{I}_d\}$, take $\mathcal{P}_{-\mathbf{I}_d} = -\mathbf{I}_d$. For $g = -\mathbf{I}_d$ (orthogonal, hence $\mathcal{P}_g^{-1} = \mathcal{P}_g^\top$):

$$\Delta_g s(\mathbf{x}) = \mathcal{P}_g^{-1} s(\mathcal{P}_g\mathbf{x}) - s(\mathbf{x}) = (-\mathcal{I})\,s(-\mathbf{x}) - s(\mathbf{x}) = -\frac{2}{\sigma^2}\,\boldsymbol{\mu}.$$

Hence

$$R(\mathbf{x}, g) = \left\|-\frac{2}{\sigma^2}\,\boldsymbol{\mu}\right\|_2^2 = \frac{4}{\sigma^4}\,\|\boldsymbol{\mu}\|_2^2, \qquad \mathbb{E}_{g\sim\nu_{\mathcal{G}}} R(\mathbf{x}, g) = \frac{2}{\sigma^4}\,\|\boldsymbol{\mu}\|_2^2,$$

since the $g = \mathbf{I}_d$ term is 0 and $\nu_{\mathcal{G}}$ is uniform. Meanwhile, $\mathbb{E}_{\mathbf{x}\sim u}\|s(\mathbf{x})\|_2^2 = d/\sigma^2$ is independent of $\boldsymbol{\mu}$. Thus, GEPC separates mean-shifts invisible to the score magnitude.

**C.2. Anisotropic covariance with $90°$ rotations ($\mathcal{G} = C_4$)**

For $d = 2$, let $\mathbf{p} = \mathcal{N}(\mathbf{0}, \boldsymbol{\Sigma})$ and $\boldsymbol{\Sigma} = \mathrm{diag}(\sigma_1^2, \sigma_2^2)$. Then $s(\mathbf{x}) = -\boldsymbol{\Sigma}^{-1}\mathbf{x}$.

Let $\mathcal{G} = C_4 = \{\mathbf{I}_d, \mathcal{R}, \mathcal{R}^2, \mathcal{R}^3\}$ where

$$\mathcal{R} = \begin{pmatrix} 0 & -1 \\ 1 & 0 \end{pmatrix}, \quad \mathcal{R}^2 = -\mathbf{I}_d, \quad \mathcal{R}^3 = \mathcal{R}^\top.$$

For any $g \in \mathcal{G}$,

$$\Delta_g s(\mathbf{x}) = \mathcal{P}_g^{-1} s(\mathcal{P}_g \mathbf{x}) - s(\mathbf{x}) = -\big(\mathcal{P}_g^\top \boldsymbol{\Sigma}^{-1} \mathcal{P}_g - \boldsymbol{\Sigma}^{-1}\big)\mathbf{x}.$$

Since $C_4$ is orthogonal, $\mathcal{P}_g^{-1} = \mathcal{P}_g^\top$.

**Compute $\mathcal{P}_g^\top \boldsymbol{\Sigma}^{-1} \mathcal{P}_g$.** For $g = \mathcal{R}$,

$$\mathcal{R}^\top \boldsymbol{\Sigma}^{-1} \mathcal{R} = \begin{pmatrix} 0 & 1 \\ -1 & 0 \end{pmatrix} \begin{pmatrix} \sigma_1^{-2} & 0 \\ 0 & \sigma_2^{-2} \end{pmatrix} \begin{pmatrix} 0 & -1 \\ 1 & 0 \end{pmatrix} = \begin{pmatrix} \sigma_2^{-2} & 0 \\ 0 & \sigma_1^{-2} \end{pmatrix}$$

Therefore

$$\mathcal{R}^\top \boldsymbol{\Sigma}^{-1} \mathcal{R} - \boldsymbol{\Sigma}^{-1} = \begin{pmatrix} \sigma_2^{-2} - \sigma_1^{-2} & 0 \\ 0 & \sigma_1^{-2} - \sigma_2^{-2} \end{pmatrix} = \big(\sigma_2^{-2} - \sigma_1^{-2}\big) \begin{pmatrix} 1 & 0 \\ 0 & -1 \end{pmatrix}.$$

Hence

$$\Delta_{\mathcal{R}} s(\mathbf{x}) = -\big(\sigma_2^{-2} - \sigma_1^{-2}\big) \begin{pmatrix} 1 & 0 \\ 0 & -1 \end{pmatrix} \mathbf{x},$$

and since $\left\| \begin{pmatrix} 1 & 0 \\ 0 & -1 \end{pmatrix} \mathbf{x} \right\|^2 = \mathbf{x}_1^2 + \mathbf{x}_2^2$,

$$R(\mathbf{x}, \mathcal{R}) := \|\Delta_{\mathcal{R}} s(\mathbf{x})\|_2^2 = \big(\sigma_2^{-2} - \sigma_1^{-2}\big)^2 \big(\mathbf{x}_1^2 + \mathbf{x}_2^2\big). \tag{44}$$

For $g = \mathcal{R}^3$, the same computation gives the same residual. For $g = \mathcal{R}^2 = -\mathbf{I}_d$, we have $(-\mathbf{I}_d)^\top \boldsymbol{\Sigma}^{-1} (-\mathbf{I}_d) = \boldsymbol{\Sigma}^{-1}$, hence $R(\mathbf{x}, \mathcal{R}^2) = 0$. Also $R(\mathbf{x}, \mathbf{I}_d) = 0$.

**Expectation under $\mathbf{x} \sim \mathcal{N}(\mathbf{0}, \boldsymbol{\Sigma})$.** We have $\mathbb{E}\big[\mathbf{x}_1^2 + \mathbf{x}_2^2\big] = \mathrm{tr}(\boldsymbol{\Sigma}) = \sigma_1^2 + \sigma_2^2$. Thus from (44),

$$\mathbb{E}_{\mathbf{x} \sim p} R(\mathbf{x}, \mathcal{R}) = \big(\sigma_2^{-2} - \sigma_1^{-2}\big)^2 \big(\sigma_1^2 + \sigma_2^2\big),$$

and averaging over $g \sim \nu_{\mathcal{G}}$ (uniform over four elements) yields

$$\mathbb{E}_{\mathbf{x} \sim p, \, g \sim \nu_{\mathcal{G}}} R(\mathbf{x}, g) = \frac{1}{2} \big(\sigma_2^{-2} - \sigma_1^{-2}\big)^2 \big(\sigma_1^2 + \sigma_2^2\big), \tag{45}$$

since only $\mathcal{R}$ and $\mathcal{R}^3$ contribute. This quantity is zero iff $\sigma_1 = \sigma_2$ (isotropy), i.e. iff the Gaussian is rotation-invariant. Hence, GEPC detects anisotropy relative to the $90°$ rotation group.

### C.3. Implementation of diffusion baselines under a shared improved-diffusion backbone

For the additional shared-backbone comparison, we re-implemented several diffusion-based OOD baselines under the same frozen CelebA-32 `improved-diffusion` checkpoint used by GEPC. These experiments are intended as *apples-to-apples test-time comparisons under a common backbone*, rather than as bit-for-bit reproductions of each method's original training and inference stack. In particular, several of the original methods were introduced in settings where the generative model is trained specifically on the in-distribution data; here, by design, we instead evaluate all methods on the same frozen backbone and adapt only their test-time scoring rule.

**Official repositories used as starting points.** When available, our rerun were initialized from the official public repositories of the corresponding methods: MSMA[1], DiffPath[2], DDPM-OOD[3], and LMD[4].

**Common backbone and notation.** Let $x_0$ denote the input image and let

$$x_t = \sqrt{\bar{\alpha}_t}\, x_0 + \sqrt{1 - \bar{\alpha}_t}\, \varepsilon, \qquad \varepsilon \sim \mathcal{N}(0, I),$$

be the corresponding noisy sample at diffusion step $t$ for the frozen `improved-diffusion` model. Depending on the baseline, we use either the model's default likelihood routine, score-related quantities derived from the predicted noise $\hat{\varepsilon}_\theta(x_t, t)$, or reconstruction errors obtained by reverse diffusion.

**NLL.** For NLL, we use the default likelihood implementation provided by `improved-diffusion`, namely the output `total_bpd` returned by `calc_bpd_loop`. This yields a per-sample negative log-likelihood proxy in bits per dimension (BPD), which we use directly as an OOD score, with larger values indicating more anomalous inputs. This baseline is therefore the closest to the original implementation.

**IC (input complexity).** For IC, we follow the complexity-corrected likelihood principle used in prior work and compute

$$s_{\mathrm{IC}}(x) = \mathrm{NLL}_{\mathrm{bpd}}(x) - C_{\mathrm{bpd}}(x),$$

where $\mathrm{NLL}_{\mathrm{bpd}}(x)$ is the DDPM likelihood score described above and $C_{\mathrm{bpd}}(x)$ is an input-complexity term obtained from lossless image compression (PNG by default, JP2 when available), normalized in bits per dimension. The normalization is required because `total_bpd` is returned in BPD whereas the raw compression length is an image-level quantity. Thus, our implementation should be interpreted as an *IC-style adaptation* to the shared improved-diffusion backbone.

**MSMA.** MSMA is implemented as a multiscale score-norm detector. We select a small set of diffusion timesteps $t_1, \ldots, t_L$ and build, for each image, the feature vector

$$\phi_{\mathrm{MSMA}}(x) = \left[ \|\hat{\varepsilon}_\theta(x_{t_1}, t_1)\|_2, \ldots, \|\hat{\varepsilon}_\theta(x_{t_L}, t_L)\|_2 \right].$$

This choice is consistent with the original MSMA intuition, since for $\varepsilon$-parameterized DDPMs the weighted score norm $\sigma_t \|s_\theta(x_t, t)\|_2$ is equivalent to $\|\hat{\varepsilon}_\theta(x_t, t)\|_2$. We then fit a Gaussian mixture model (GMM) on ID-train features and use the negative log-density under this GMM as the OOD score. Hence, this baseline is best viewed as an *MSMA-style adaptation* under discrete DDPM timesteps and a shared frozen backbone.

**DDPM-OOD.** DDPM-OOD is implemented as a reconstruction-based detector. We first define a DDIM inference grid and select several starting timesteps. For each selected start, we reconstruct the image by deterministic reverse diffusion and compute the reconstruction MSE with respect to the input. Following the logic of the original method, we calibrate these reconstruction errors using ID-train statistics on a per-start basis and aggregate the resulting standardized scores across starts. Because the original implementation relies on a different inference stack, our version should be interpreted as a *DDPM-OOD-style reconstruction baseline adapted to improved-diffusion*.

**LMD.** LMD is implemented as a masked reconstruction detector. For each image, we hide part of the input using a fixed masking pattern (checkerboard by default), perform masked reverse diffusion / inpainting with the frozen backbone, and compare the reconstructed image to the original one using MSE. This process is repeated multiple times per image, and the final score is obtained by median aggregation across repetitions. This preserves the core idea of LMD—detecting OOD samples through masked reconstruction inconsistency—while adapting the reconstruction mechanism to the shared improved-diffusion backbone. Our implementation should therefore be understood as an *LMD-style adaptation* rather than an exact reproduction of the original score-SDE-based stack.

---

[1] https://github.com/ahsanMah/msma
[2] https://github.com/clear-nus/diffpath
[3] https://github.com/marksgraham/ddpm-ood
[4] https://github.com/zhenzhel/lift_map_detect

**Interpretation.** Overall, these baselines serve two purposes. First, they preserve the main intuition of the original methods. Second, they allow a strictly shared-backbone comparison in which methods differ only through their test-time statistic or reconstruction rule. This isolates the contribution of the OOD score itself and complements the broader benchmark numbers reported from prior work under their original protocols.

## D. Experimental details and reproducibility

**Implementation.** For GEPC, we follow the ID-only protocol of Section 6: ID-train is used for timestep selection, weighting, and density calibration; ID-test and OOD-test are used only for evaluation.

**Hardware and software.** Unless stated otherwise, experiments are run on a single GPU (NVIDIA GeForce RTX 4060 Laptop GPU) with PyTorch on Linux.

**Determinism.** We fix seeds for Python, NumPy, and PyTorch, disable TF32, and optionally enable PyTorch deterministic algorithms. DataLoaders use an explicit `torch.Generator` with a fixed seed and `worker_init_fn` to ensure stable shuffling across workers. We report exact command lines and YAML configs in the released code.

**Compute accounting.** We report compute as F+J, where $F$ is a forward evaluation of $\mathbf{s}_\theta(\cdot, t)$ and $J$ is a Jacobian–vector product counted as a forward-equivalent operation. For methods using $T$ reverse diffusion steps, we report the corresponding number of sequential score evaluations.

## E. GEPC feature variants and fusion

Let $\mathbf{x}_t \sim q(\mathbf{x}_t \mid \mathbf{x}_0)$. Define the transported score residual field

$$\mathbf{r}_t(\mathbf{x}_t, g) := \mathcal{P}_g^{-1}\mathbf{s}_\theta(\mathcal{P}_g\mathbf{x}_t, t) - \mathbf{s}_\theta(\mathbf{x}_t, t) \in \mathbb{R}^{C \times h \times w}, \tag{46}$$

and the transported score in the canonical frame

$$\tilde{\mathbf{s}}_\theta(\mathbf{x}_t, t; g) := \mathcal{P}_g^{-1}\mathbf{s}_\theta(\mathcal{P}_g\mathbf{x}_t, t) \in \mathbb{R}^{C \times h \times w}, \tag{47}$$

so that $\mathbf{r}_t(\mathbf{x}_t, g) = \tilde{\mathbf{s}}_\theta(\mathbf{x}_t, t; g) - \mathbf{s}_\theta(\mathbf{x}_t, t)$. Throughout, $\mathrm{pool}(\cdot)$ denotes the following convention: for $A \in \mathbb{R}^{C \times h \times w}$ we first average over channels and then pool over spatial locations by either mean-pooling or top-$k$ pooling (top-$k$ averages the $k$ largest spatial responses). We apply this to pointwise energies, e.g. $\mathrm{pool}(\|\mathbf{u}\|_2^2)$.

**Baseline normaliser.** We use the pooled score energy

$$b_t(\mathbf{x}_0) := \mathrm{pool}\Big(\|\mathbf{s}_\theta(\mathbf{x}_t, t)\|_2^2\Big). \tag{48}$$

**GEPC$_s$ (base-normalised residual energy).**

$$z_t^{(s)}(\mathbf{x}_0) := \mathbb{E}_{g \sim \mathrm{Unif}(\mathcal{G})}\left[\frac{\mathrm{pool}\Big(\|\mathbf{r}_t(\mathbf{x}_t, g)\|_2^2\Big)}{b_t(\mathbf{x}_0)}\right]. \tag{49}$$

**GEPC$_{\mathrm{cos}}$ (global cosine inconsistency).** Let $\langle a, b \rangle$ denote the dot product after vectorising over $(c, h, w)$, and $\|a\|$ the corresponding Euclidean norm. We use

$$z_t^{(\mathrm{cos})}(\mathbf{x}_0) := \mathbb{E}_{g \sim \mathrm{Unif}(\mathcal{G})}\left[1 - \frac{\langle \tilde{\mathbf{s}}_\theta(\mathbf{x}_t, t; g), \mathbf{s}_\theta(\mathbf{x}_t, t) \rangle}{\|\tilde{\mathbf{s}}_\theta(\mathbf{x}_t, t; g)\| \, \|\mathbf{s}_\theta(\mathbf{x}_t, t)\|}\right], \tag{50}$$

which is scale-invariant and thus requires no additional base normalisation.

**GEPC$_{\mathrm{pair}}$ (pairwise dispersion, base-normalised).** We also use explicit pair enumeration:

$$z_t^{(\mathrm{pair})}(\mathbf{x}_0) := \mathbb{E}_{g < g'}\left[\frac{\mathrm{pool}\Big(\|\tilde{\mathbf{s}}_\theta(\mathbf{x}_t, t; g) - \tilde{\mathbf{s}}_\theta(\mathbf{x}_t, t; g')\|_2^2\Big)}{b_t(\mathbf{x}_0)}\right]. \tag{51}$$

---

**Algorithm 1** Stability-Based Timestep Selection (ID-only)

---

1: **Input:** ID-train set $\mathcal{X}$, candidate timesteps $\mathcal{T}_{\text{cand}}$, integer $K$
2: **Output:** selected timesteps $\mathcal{T}$ and weights $\{w_t\}$
3: **for** $t \in \mathcal{T}_{\text{cand}}$ **do**
4:     Compute scores $\{z_t(\mathbf{x})\}_{\mathbf{x} \in \mathcal{X}}$ (default: $z_t^{(s)}(\mathbf{x})$)
5:     $\text{CV}(t) \leftarrow \text{std}(z_t)/(|\text{mean}(z_t)|)$
6: **end for**
7: $\mathcal{T} \leftarrow$ the $K$ timesteps with smallest $\text{CV}(t)$
8: $w_t \propto 1/(\text{CV}(t))$ for $t \in \mathcal{T}$ and normalise $\sum_{t \in \mathcal{T}} w_t = 1$
9: **return** $\mathcal{T}$ and $\{w_t\}$

---

**ID-only calibration and fusion.**  Let $\mathcal{F} = \{s, \cos, \text{pair}\}$ denote the enabled feature set. In the default scalar-density mode (`vector_mode=none`), we fit an ID-only model per $(t, f)$ on ID-train: (i) KDE (`density_mode=kde`) provides $\log p_{t,f}(z)$, (ii) z-score (`density_mode=zscore`) provides $\ell_{t,f}(z) = -\frac{1}{2}((z - \mu_{t,f})/\sigma_{t,f})^2$, or (iii) raw (`density_mode=none`) uses $z$ directly. Within a timestep, we aggregate per-feature scores using `agg_feat` (sum/mean), then aggregate across timesteps using `agg_t` (default: inverse-CV weighted mean). For KDE/z-score, the ID score is

$$L(\mathbf{x}_0) \coloneqq \sum_{t \in \mathcal{T}} w_t \, \text{AggFeat}\big(\{\ell_{t,f}(z_t^{(f)}(\mathbf{x}_0))\}_{f \in \mathcal{F}}\big), \tag{52}$$

and the final anomaly score is $S(\mathbf{x}_0) \coloneqq -L(\mathbf{x}_0)$ (OOD-high). In raw mode, we directly set $S(\mathbf{x}_0)$ to the corresponding aggregated one-sided statistic.

**Vector MVN (optional).**  In `vector_mode=mvn`, we fit a single Gaussian on the concatenated feature vector over all kept $(t, f)$ on ID-train and score with the corresponding Mahalanobis distance (OOD-high). Importantly, all three features reuse the same score-network evaluations, so enabling multiple features does not change the NFE.

**Timestep Selection Algorithm.**

# F. Additional ablations and runtime

This appendix reports comprehensive ablations for GEPC on the $32 \times 32$ setting. Unless stated otherwise, ablations follow the default configuration in Section 6.1 and are reported for **all 9 ID/OOD pairs**. For readability, we additionally provide representative plots for one pair (SVHN as ID, CIFAR-100 as OOD) in Figs. 3–4.

### F.1. SNR-to-timestep mapping

For DDPM-style schedules, we use $\text{SNR}(t) \coloneqq \bar{\alpha}_t/(1 - \bar{\alpha}_t)$ and map each target SNR level (`snr_levels`) to the closest discrete index $t$ by nearest-neighbour matching on the precomputed schedule. This yields a small candidate set $\mathcal{T}_{\text{cand}}$.

### F.2. ID-only timestep selection and weighting

For each $t \in \mathcal{T}_{\text{cand}}$, we compute an ID-only stability score using the coefficient of variation

$$\text{CV}(t) = \frac{\text{std}(z_t(\mathbf{x}))}{|\text{mean}(z_t(\mathbf{x}))|},$$

over ID-train samples (default: $z_t^{(s)}$). We keep the $K$ most stable timesteps (lowest CV), yielding $\mathcal{T}$, and set

$$w_t \propto \frac{1}{\text{CV}(t)},$$

(`weight_t=inv_cv`), normalised to sum to one. We use `agg_t=wmean` unless stated otherwise, and fix $K$ across datasets in the main table to keep compute comparable.

*Table 3.* Timestep candidates, selected timesteps, and (kept-only normalised) weights for the default configuration ($K = 2$, `weight_t=inv_cv`).

| | CIFAR-10 (ID) | | | SVHN (ID) | | | CelebA (ID) | | |
| --- | --- | --- | --- | --- | --- | --- | --- | --- | --- |
| | vs SVHN | vs CelebA | vs C100 | vs C10 | vs CelebA | vs C100 | vs C10 | vs SVHN | vs C100 |
| $\mathcal{T}_{\text{cand}}$ | $\{5, 15, 136, 172\}$ | $\{5, 15, 136, 172\}$ | $\{5, 15, 136, 172\}$ | $\{5, 15, 136, 172\}$ | $\{5, 15, 136, 172\}$ | $\{5, 15, 136, 172\}$ | $\{5, 86, 172, 332\}$ | $\{5, 86, 172, 332\}$ | $\{5, 86, 172, 332\}$ |
| kept $\mathcal{T}$ ($K = 2$) | $\{5, 136\}$ | $\{5, 136\}$ | $\{5, 136\}$ | $\{5, 15\}$ | $\{5, 15\}$ | $\{5, 15\}$ | $\{86, 172\}$ | $\{86, 172\}$ | $\{86, 172\}$ |
| weights on kept | $(0.520, 0.480)$ | $(0.520, 0.480)$ | $(0.522, 0.478)$ | $(0.429, 0.571)$ | $(0.429, 0.571)$ | $(0.428, 0.572)$ | $(0.501, 0.499)$ | $(0.501, 0.499)$ | $(0.502, 0.498)$ |

*Table 4.* Timestep selection sweep across 9 ID/OOD pairs. We report AUROC and the implied NFE per input ($8K$).

| $K$ | $w_t$ | NFE/img | CIFAR-10 (ID) | | | SVHN (ID) | | | CelebA (ID) | | |
| --- | --- | --- | --- | --- | --- | --- | --- | --- | --- | --- | --- |
| | | | vs SVHN | vs CelebA | vs C100 | vs C10 | vs CelebA | vs C100 | vs C10 | vs SVHN | vs C100 |
| 1 | none | 8 | 0.871 | 0.933 | 0.534 | 0.756 | 1.000 | 0.799 | 0.999 | 0.999 | 0.999 |
| 2 | none | 16 | 0.835 | 0.999 | 0.554 | 0.891 | 1.000 | 0.903 | 1.000 | 1.000 | 1.000 |
| 3 | none | 24 | 0.785 | 0.999 | 0.554 | 0.865 | 1.000 | 0.884 | 1.000 | 1.000 | 1.000 |
| 4 | none | 32 | 0.758 | 0.999 | 0.565 | 0.842 | 1.000 | 0.864 | 1.000 | 1.000 | 0.999 |
| 1 | inv_cv | 8 | 0.870 | 0.933 | 0.539 | 0.760 | 1.000 | 0.801 | 1.000 | 1.000 | 0.999 |
| 2 | inv_cv | 16 | 0.841 | 0.999 | 0.558 | 0.879 | 1.000 | 0.894 | 1.000 | 1.000 | 1.000 |
| 3 | inv_cv | 24 | 0.791 | 0.999 | 0.556 | 0.863 | 1.000 | 0.880 | 1.000 | 1.000 | 0.999 |
| 4 | inv_cv | 32 | 0.769 | 0.999 | 0.566 | 0.845 | 1.000 | 0.868 | 1.000 | 1.000 | 0.999 |

**Two per-$t$ diagnostics.** We distinguish (i) a component-level diagnostic that reports AUROC of the raw transported gap at each single timestep (Figure 3c), and (ii) the AUROC of the final GEPC score when evaluated using a *single* timestep (stored alongside the $K$-sweep in Table 4). The former explains *where* symmetry-breaking arises; the latter supports the ID-only selection rule.

**Selected timesteps and weights (9 pairs).** Table 3 reports $\mathcal{T}_{\text{cand}}$, the default kept set ($K = 2$, inv_cv), and the corresponding weights (normalised over kept timesteps).

**Sweep over $K$ and weighting (all 9 pairs).** Table 4 reports a sweep over $K \in \{1, 2, 3, 4\}$ and weighting choices for *all* 9 ID/OOD pairs. We include the implied NFE per input ($= (1 + |\mathcal{G}|) K = 8K$ with $|\mathcal{G}| = 7$).

**Sweep over $K$ and weighting (representative pair).** For direct comparison with the plots in Figs. 3–4, Table 5 reports the same sweep for SVHN (ID) vs CIFAR-100 (OOD).

### F.3. Sensitivity to the choice of transformation group $G$

GEPC relies on approximate score-field equivariance on ID data under a finite transformation group $G$. To assess sensitivity to this design choice, we perform an ablation on representative $32 \times 32$ benchmarks with two different ID datasets, namely **SVHN** and **CIFAR10**, and three OOD datasets for each ID (**CIFAR10**, **CelebA**, and **CIFAR100** for SVHN; **SVHN**, **CelebA**, and **CIFAR100** for CIFAR10), as shown in Table 6. Concretely, our default group contains horizontal/vertical flips, $90°/180°$ rotations, and 1-pixel circular shifts. We compare the full group against three single-family subgroups (*flips only*, *rotations only*, *shifts only*) and three leave-one-family-out variants (*no flips*, *no rotations*, *no shifts*).

To isolate the effect of the group choice itself, we keep the checkpoint, preprocessing, ID/OOD splits, timestep selection, calibration, and all other hyperparameters fixed, and vary only the transformation group. In addition to AUROC/AUPR/FPR95, we report an ID-only diagnostic, namely the coefficient of variation (CV) of GEPC scores on held-out ID data. Intuitively, a mismatched group increases the residual floor already on ID data, which tends to increase ID score variability and reduce the ID/OOD margin.

The ablation in Table 6 shows that GEPC is sensitive to the choice of $G$, but in a structured and interpretable way rather than an arbitrary one. Across both ID datasets, the *shifts only* variant is consistently the weakest or among the weakest, and is also associated with the largest ID-CV, indicating that an unstable equivariance signal already appears on held-out ID data. By contrast, simpler groups without shifts, such as *flips only*, *rotations only*, or *no shifts*, are consistently more stable and often achieve better OOD performance. In particular, for SVHN, *flips only* performs best on average, while for CIFAR10, *no shifts* achieves the best average AUROC/AUPR and the lowest ID-CV.

These results suggest that not all transformation families contribute equally across domains. In both SVHN and CIFAR10,

*Table 5.* Timestep selection sweep (SVHN as ID, CIFAR-100 as OOD). We report AUROC and predicted NFE per input ($= 8K$). Best is **bold**, second best is underlined.

| $K$ | weighting $w_t$ | AUROC | NFE/img |
|---|---|---|---|
| 1 | none | 0.799 | 8 |
| 2 | none | **0.903** | 16 |
| 3 | none | 0.884 | 24 |
| 4 | none | 0.864 | 32 |
| 1 | inv_cv | 0.801 | 8 |
| 2 | inv_cv | 0.894 | 16 |
| 3 | inv_cv | 0.880 | 24 |
| 4 | inv_cv | 0.868 | 32 |

*Table 6.* Sensitivity of GEPC to the choice of transformation group $G$ on 32×32 benchmarks with two different ID datasets. For each ID, AUROC, AUPR, and FPR95 are averaged over the three corresponding OOD datasets. Lower ID-CV indicates a more stable equivariance signal on held-out ID data.

| (a) SVHN as ID | | | | | (b) CIFAR10 as ID | | | | |
|---|---|---|---|---|---|---|---|---|---|
| **Group** | **AUROC** | **AUPR** | **FPR95** | **ID-CV** | **Group** | **AUROC** | **AUPR** | **FPR95** | **ID-CV** |
| Full group | 0.925 | 0.933 | 0.351 | 0.239 | Full group | 0.798 | 0.775 | **0.466** | 0.222 |
| Flips only | **0.945** | **0.954** | **0.300** | **0.143** | Flips only | 0.786 | 0.766 | 0.493 | 0.285 |
| Rotations only | 0.934 | 0.946 | 0.350 | 0.154 | Rotations only | 0.793 | 0.770 | 0.484 | 0.239 |
| Shifts only | 0.869 | 0.875 | 0.495 | 1.124 | Shifts only | 0.754 | 0.721 | 0.561 | 0.316 |
| No flips | 0.915 | 0.921 | 0.395 | 0.397 | No flips | 0.793 | 0.766 | 0.476 | 0.267 |
| No rotations | 0.912 | 0.918 | 0.395 | 0.461 | No rotations | 0.782 | 0.754 | 0.488 | 0.286 |
| No shifts | 0.937 | 0.949 | 0.331 | 0.160 | No shifts | **0.802** | **0.785** | 0.475 | **0.197** |

which have a relatively canonical orientation and centered content, circular shifts appear less appropriate than flips or rotations. For a new domain, we therefore recommend selecting $G$ from a small set of plausible candidate groups using held-out ID data only. In practice, one should prefer groups that yield both a low ID residual floor and a low ID score variability (e.g., low held-out ID-CV), while excluding transformations that clearly violate the geometry or semantics of the domain.

### F.4. Sensitivity to Monte Carlo forward noise

To assess the variability induced by the Gaussian noise $\epsilon$ used in the forward process, we keep the checkpoint, preprocessing, ID/OOD splits, transformation group $G$, timestep selection, calibration, and all hyperparameters fixed, and vary only the random seed controlling the Monte Carlo noise. Concretely, we fit GEPC once on the same ID training split and re-score the same ID/OOD test sets over 5 independent seeds. Table 7 reports the resulting mean and standard deviation across seeds for three representative 32×32 benchmark pairs with SVHN as ID and CIFAR10, CelebA, and CIFAR100 as OOD. This isolates the effect of stochastic forward noise rather than changes in data splits or hyperparameters.

The observed standard deviations in Table 7 are very small across all three pairs, indicating that GEPC is stable with respect to Monte Carlo forward noise under the reported configuration. In particular, the variance remains negligible both on an easier pair (SVHN → CelebA) and on more moderate-difficulty pairs (SVHN → CIFAR10 and SVHN → CIFAR100).

### F.5. Complement to pairwise AUROC results

To complement the pairwise AUROC results, we additionally report average AUPR and average FPR95 over the same 9 standard $32 \times 32$ ID/OOD pairs under the shared-backbone CelebA-32 protocol in table F.5. This appendix table is intentionally compact: its goal is to summarize the broader operating characteristics of the methods under a common frozen backbone, while the main shared-backbone table reports the detailed pairwise AUROC values. In all cases, OOD is treated as the positive class and higher scores indicate more anomalous samples; therefore, higher AUPR is better whereas lower FPR95 is better.

*Table 7.* Sensitivity of GEPC to Monte Carlo forward noise on representative 32×32 benchmark pairs. We keep the fitted GEPC model and all evaluation settings fixed, and vary only the random seed controlling the Gaussian noise $\epsilon$ in the forward process. Results are reported as mean $\pm$ standard deviation over 5 seeds.

| Pair | AUROC ↑ | AUPR ↑ | FPR95 ↓ |
|---|---|---|---|
| SVHN → CIFAR10 | $0.8802 \pm 0.0009$ | $0.8918 \pm 0.0005$ | $0.5442 \pm 0.0073$ |
| SVHN → CelebA | $0.9999 \pm 0.0000$ | $0.9999 \pm 0.0000$ | $0.0000 \pm 0.0000$ |
| SVHN → CIFAR100 | $0.8948 \pm 0.0010$ | $0.9078 \pm 0.0011$ | $0.5104 \pm 0.0132$ |

*Table 8.* Shared-backbone comparison on a single frozen CelebA-32 improved-diffusion checkpoint. We report the average AUROC, average AUPR, and average FPR95 over the 9 standard $32 \times 32$ ID/OOD pairs. Higher is better for AUROC and AUPR; lower is better for FPR95. All methods are evaluated under the same frozen backbone and differ only by their test-time statistic or reconstruction rule. Compute is reported as $F + J$ (forward passes + JVPs).

| Method | Avg. AUROC ↑ | Avg. AUPR ↑ | Avg. FPR95 ↓ | $F + J$ |
|---|---|---|---|---|
| NLL | 0.581 | 0.661 | 0.807 | $4000F + 0J$ |
| IC (diffusion) | 0.536 | 0.613 | 0.740 | $4000F + 0J$ |
| MSMA | 0.861 | 0.864 | 0.481 | $10F + 0J$ |
| DDPM-OOD | 0.586 | 0.647 | 0.750 | $364F + 0J$ |
| LMD | 0.515 | 0.538 | 0.854 | $10^4 F + 0J$ |
| **GEPC_lite (ours)** | **0.910** | **0.907** | **0.265** | $8F + 0J$ |
| **GEPC (ours)** | 0.908 | 0.904 | 0.274 | $16F + 0J$ |

## F.6. OpenOOD-style large-scale

We emphasize that Table 9 is intended as a matched diffusion-side scalability check rather than a direct comparison to supervised classifier-based OpenOOD baselines. Its role is to assess whether the GEPC equivariance-consistency signal remains informative beyond the 32×32 CelebA setting, on a larger unconditional ImageNet-64 diffusion backbone and standard ImageNet-style OOD datasets.

## F.7. Per-transform ablation (group elements)

Let $\mathcal{G}$ denote the set of transported inputs used by GEPC. We compute an AUROC for each $g \in \mathcal{G}$ by isolating the corresponding group-consistency gap, and compare it to the AUROC obtained by averaging over all transforms. Figure 3 (middle) shows a representative example.

**What is varied in the per-$g$ plot.** For interpretability, per-transform AUROCs are computed from the *raw* transported-gap component (i.e. without KDE/z-score calibration), averaged over the retained timesteps. The dashed horizontal line corresponds to averaging the same raw gap over all $g \in \mathcal{G}$ ("mean over $g$" in Figure 3b). This diagnostic checks that performance is not driven by a single transform.

**9-pair summary table.** Table 10 summarises the AUROC obtained by averaging the raw gap over $g \in \mathcal{G}$. Since this diagnostic is *unsigned* (the raw gap can be ID-high or OOD-high depending on the pair), we report $\max(\mathrm{AUROC}, 1 - \mathrm{AUROC})$ as a sign-invariant separability score.

## F.8. Calibration variants and feature fusion

We compare KDE calibration (`density_mode=kde`) against z-score normalisation and the uncalibrated score (`density_mode=none`). We also evaluate a Gaussian/Mahalanobis model on multi-$t$ feature vectors (`vector_mode=mvn`).

**Calibration variants (9 pairs).** Table 11 reports AUROC for calibration choices using the single feature $\mathrm{GEPC}_s$.

## F.9. Feature variants (single-feature ablations)

We ablate the three GEPC statistics used in the paper (Appendix E for definitions). For compactness, Table 12 reports the single-feature AUROC for each statistic across 9 pairs. Figure 3 (left) visualises a representative case.

*Table 9.* OpenOOD-style large-scale stress test on an unconditional ImageNet-64 diffusion backbone. We compare GEPC and MSMA under a matched checkpoint, preprocessing, and ID/OOD protocol. ImageNet-64 is used as ID; Textures is far-OOD, while SSB-hard is near-OOD. This is a diffusion-side comparison and not an apples-to-apples comparison against supervised classifier-based post-hoc baselines.

| Method | SSB-hard | Textures | Mean |
|--------|----------|----------|------|
| MSMA   | 0.591    | 0.718    | 0.655 |
| GEPC   | 0.640    | 0.718    | 0.679 |

*Table 10.* Per-transform ablation summary (9 ID/OOD pairs). We report sign-invariant AUROC of the group-averaged raw statistic: $\max(\text{AUROC}, 1 - \text{AUROC})$.

| | CIFAR-10 (ID) | | | SVHN (ID) | | | CelebA (ID) | | |
|---|---|---|---|---|---|---|---|---|---|
| Metric | vs SVHN | vs CelebA | vs C100 | vs C10 | vs CelebA | vs C100 | vs C10 | vs SVHN | vs C100 |
| $\max(\text{AUROC}, 1 - \text{AUROC})$ | 0.858 | 0.999 | 0.539 | 0.915 | 1.000 | 0.923 | 1.000 | 1.000 | 1.000 |

### F.10. Runtime and NFEs

For each timestep $t$, GEPC uses one reference evaluation $\mathbf{s}_\theta(\mathbf{x}_t, t)$ and one batched evaluation over transported inputs $\{\mathcal{P}_g \mathbf{x}_t\}_{g \in \mathcal{G}}$, hence $(1 + |\mathcal{G}|)$ forward evaluations and 0 JVPs per timestep. With $m$ Monte-Carlo noise samples and $K = |\mathcal{T}|$ retained timesteps, total cost is $(1 + |\mathcal{G}|) K m$ forward passes. This computation is parallelisable over $g$ and (when memory allows) over $t$.

### F.11. Representative plots and score distributions

We provide representative plots for one pair (CIFAR-10 as ID, SVHN as OOD). Figure 3 shows feature variants, per-transform AUROC, and single-timestep AUROC vs. $t$. Figure 4 shows the separation of score distributions for baseline energy, transported energy gap, and the final GEPC$_s$ score.

## G. Radar SAR details

**SAR background (context).** Synthetic Aperture Radar (SAR) is an active microwave imaging modality producing high-resolution reflectivity maps under all-weather and day/night conditions. SAR images are coherent and typically exhibit speckle and strong intensity dynamics; we therefore visualise and process patches in log-magnitude.

**Datasets and OOD task.** We use HRSID and SSDD, two public SAR datasets commonly used for ship detection. We form an OOD task where *sea-clutter-only* patches are in-distribution (ID) and patches containing at least one annotated ship (and wake when visible in the patch) are out-of-distribution (OOD).

**Quantitative results.** Patch-level OOD detection metrics are reported in Table 14 for ID sea-clutter patches from HRSID against target-containing patches from HRSID (intra-dataset) and SSDD (cross-dataset).

We additionally report the classical Reed–Xiaoli (RX) detector on the SAR experiment. RX is fitted on clutter-only ID-train patches and evaluated on the same ID/OOD SAR splits as GEPC. Because the SAR patches are high-dimensional, we apply RX in a PCA-reduced space with regularized covariance estimation. Concretely, each patch is mapped to a low-dimensional feature vector $z$, and scored by the squared Mahalanobis distance

$$s_{\text{RX}}(x) = (z - \mu)^\top \Sigma^{-1} (z - \mu),$$

with $(\mu, \Sigma)$ estimated from ID-train only. Higher scores indicate stronger OOD evidence. We report AUROC, AUPR, and FPR95 for direct comparison with GEPC.

**Preprocessing and patching.** For each SAR patch, we convert intensities to log-magnitude, apply per-patch normalisation, and resize/crop to $256 \times 256$ to match the LSUN-256 diffusion backbone input. If the backbone expects 3 channels, we replicate the single-channel SAR patch across channels. No SAR-specific fine-tuning is performed.

*Table 11.* Calibration variants for GEPC$_s$ across 9 ID/OOD pairs. Values are AUROC. Best is **bold**, second best is underlined *within each column*.

| | CIFAR-10 (ID) | | | SVHN (ID) | | | CelebA (ID) | | |
|---|---|---|---|---|---|---|---|---|---|
| Calibration | vs SVHN | vs CelebA | vs C100 | vs C10 | vs CelebA | vs C100 | vs C10 | vs SVHN | vs C100 |
| KDE (ID-only) | 0.840 | **0.999** | 0.556 | 0.879 | **1.000** | 0.894 | **1.000** | **1.000** | **1.000** |
| z-score | **0.841** | **0.999** | 0.557 | 0.854 | **1.000** | 0.873 | **1.000** | **1.000** | **1.000** |
| none (raw) | 0.136 | **0.999** | 0.538 | **0.911** | **1.000** | **0.918** | 0.000 | 0.000 | 0.000 |
| MVN (Mahalanobis) | 0.838 | **0.999** | **0.559** | 0.881 | **1.000** | 0.891 | 0.999 | **1.000** | 0.999 |

*Table 12.* Single-feature ablations across 9 ID/OOD pairs (three GEPC statistics). Values are AUROC under KDE calibration. Best is **bold**, second best is underlined *within each column*.

| | CIFAR-10 (ID) | | | SVHN (ID) | | | CelebA (ID) | | |
|---|---|---|---|---|---|---|---|---|---|
| Feature | vs SVHN | vs CelebA | vs C100 | vs C10 | vs CelebA | vs C100 | vs C10 | vs SVHN | vs C100 |
| GEPC$_s$ | **0.839** | **0.999** | 0.556 | **0.879** | **1.000** | **0.896** | **1.000** | **1.000** | **0.999** |
| GEPC$_{\text{cos}}$ | 0.584 | **0.999** | 0.546 | 0.873 | **1.000** | **0.896** | **1.000** | **1.000** | **0.999** |
| GEPC$_{\text{pair}}$ | 0.820 | 0.997 | 0.550 | 0.861 | 0.999 | 0.877 | 0.999 | 0.999 | 0.998 |
| Fusion (mean) | 0.831 | **0.999** | **0.557** | 0.876 | **1.000** | 0.894 | **1.000** | **1.000** | **0.999** |

**Equivariance residual maps and normalisation.** Beyond the scalar GEPC score, we visualise the *pre-pooling* equivariance residual magnitude map $|\Delta(\mathbf{x})|$, highlighting spatial regions where equivariance breaks (typically ships/wakes) while remaining low on homogeneous sea clutter. For magnitude comparison across examples and datasets, we export globally normalised maps using a fixed $v_{\text{global}} = \text{median}_{\mathbf{x} \in \mathcal{P}_{\text{ID}}} q_{0.99}(|\Delta(\mathbf{x})|)$, computed over an ID candidate pool $\mathcal{P}_{\text{ID}}$. We also export per-image normalised maps and raw residual maps for inspection (see exported files and metadata).

*Table 13.* Measured runtime for a representative $32 \times 32$ pair (SVHN as ID, CIFAR-100 as OOD) on a single GPU, alongside implied NFE. Timing is reported as milliseconds per image (lower is better). Hardware: NVIDIA GeForce RTX 4060 Laptop GPU (Linux, PyTorch).

| Variant | AUROC | ms/img (ID) | ms/img (OOD) |
|---|---|---|---|
| GEPC$_s$ + KDE | 0.894 | 69.920 | 69.960 |
| GEPC$_s$ + z-score | 0.873 | 70.020 | 69.720 |
| GEPC$_s$ raw | 0.918 | 69.680 | 69.660 |
| GEPC$_s$ + MVN | 0.891 | 69.860 | 69.690 |

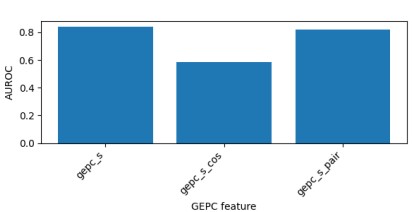

*(a)* Feature variants (single-feature AU-ROC).

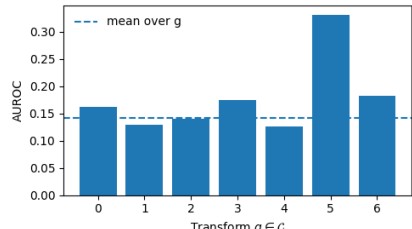

*(b)* Per-transform AUROC (raw gap component).

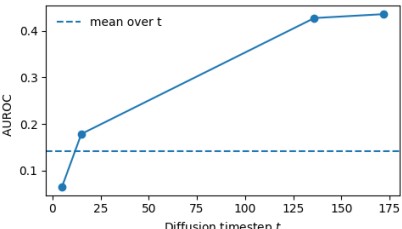

*(c)* Single-timestep AUROC vs. $t$ (raw gap component).

*Figure 3.* Representative ablations for GEPC (CIFAR10 as ID, SVHN as OOD). (a) Single-feature variants under the same ID-only protocol. (b) Per-transform AUROC computed from the raw transported-gap component (no calibration); the dashed line averages the same component over $g \in \mathcal{G}$. (c) Single-timestep AUROC computed from the raw transported-gap component; the dashed line averages the same component over the retained timesteps.

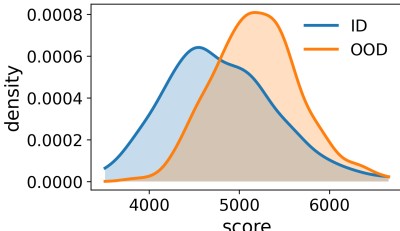

*(a)* **Score magnitude (non-GEPC).** $E_t(\mathbf{x}_t) := \|\mathbf{s}_\theta(\mathbf{x}_t, t)\|_2^2$, with $\mathbf{x}_t \sim q(\cdot \mid \mathbf{x}_0)$.

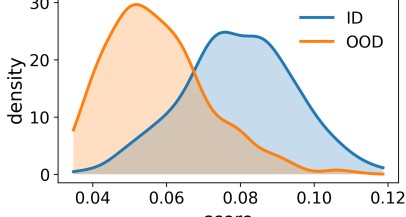

*(b)* **Equivariance residual energy (single-step).** $R_t(\mathbf{x}_t, g) := \|\Delta_g \mathbf{s}_\theta(\mathbf{x}_t, t)\|_2^2$, where $\Delta_g f(\mathbf{x}, t) := \mathcal{P}_g^{-1} f(\mathcal{P}_g \mathbf{x}, t) - f(\mathbf{x}, t)$.

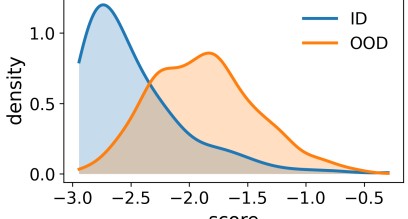

*(c)* **Final GEPC score (time-averaged).** $\text{GEPC}(\mathbf{x}_0) := \sum_{t \in \mathcal{T}} w_t \, \mathbb{E}_{\mathbf{x}_t \sim q(\cdot|\mathbf{x}_0), \, g \sim \nu_\mathcal{G}}[R_t(\mathbf{x}_t, g)].$

*Figure 4.* **Score distributions (ID vs OOD) for a representative pair (SVHN as ID, CIFAR-100 as OOD).** Left: score magnitude $E_t(\mathbf{x}_t)$ (a baseline diagnostic, not GEPC). Middle: single-step equivariance residual energy $R_t(\mathbf{x}_t, g)$. Right: time-averaged GEPC score $\text{GEPC}(\mathbf{x}_0)$ aggregating $R_t$ over $t \in \mathcal{T}$ with weights $w_t$ and uniform $g \sim \nu_\mathcal{G}$.

*Table 14.* Patch-level OOD detection on SAR. ID is sea-clutter patches from HRSID; OOD are target-containing patches from HRSID and SSDD. Higher AUROC/AUPR is better; lower FPR@95%TPR is better.

| OOD split (targets) | Method | AUROC ↑ | FPR@95 ↓ | AUPR ↑ |
|---|---|---|---|---|
| HRSID-ship/wake | RX | 0.573 | 0.613 | 0.552 |
| HRSID-ship/wake | GEPC | 0.853 | 0.780 | 0.777 |
| SSDD-ship | RX | 1.000 | 0.000 | 1.000 |
| SSDD-ship | GEPC | 1.000 | 0.000 | 1.000 |

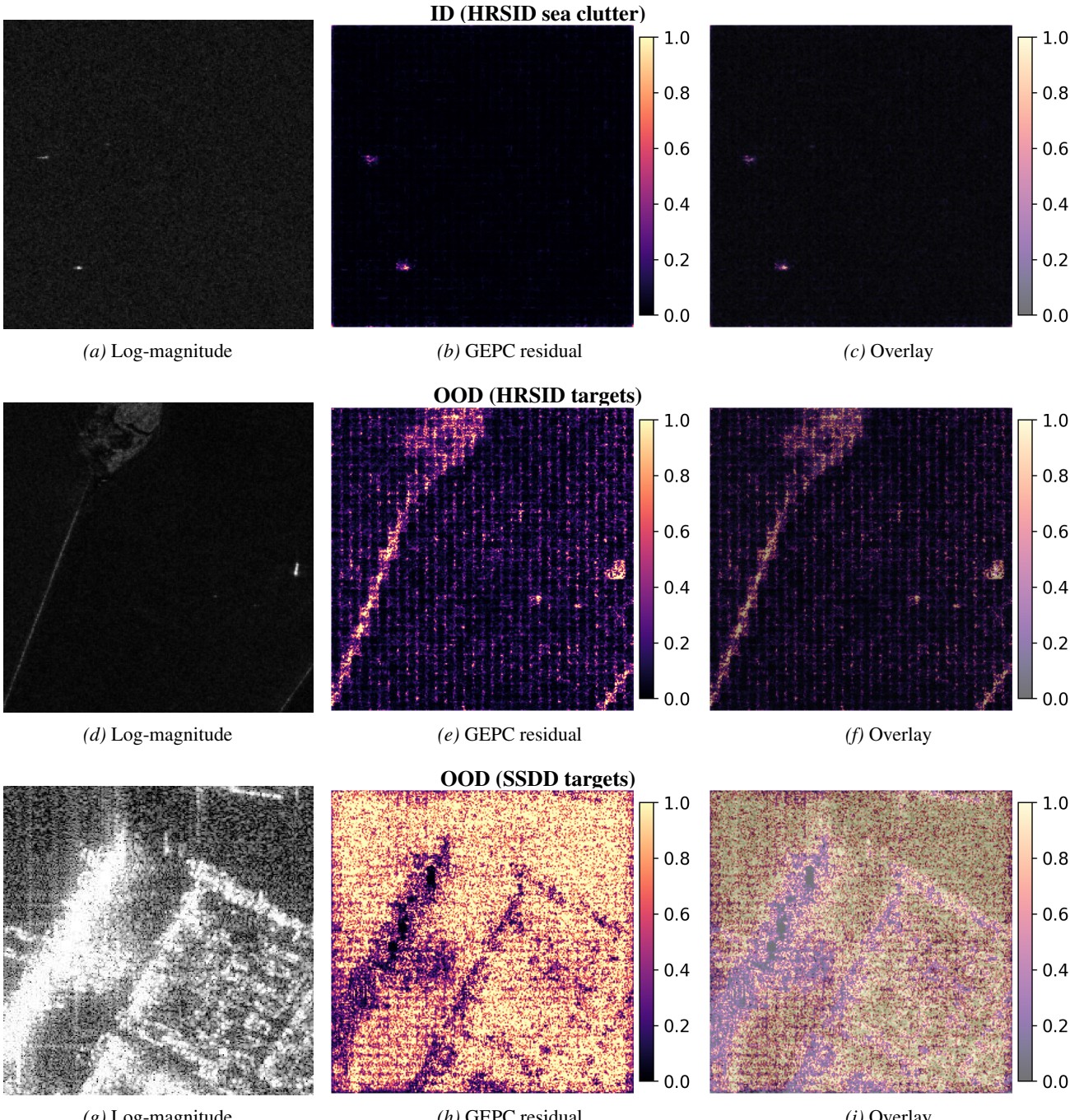

*Figure 5.* Qualitative GEPC localisation on SAR patches (LSUN-256, no SAR fine-tuning). Residual maps are globally normalised by a shared $v_{\text{global}}$ (computed on an ID pool) to enable comparison across ID/OOD and across datasets.

