# OpenReview forum: "GEPC: Group-Equivariant Posterior Consistency for Out-of-Distribution Detection in Diffusion Models"
_ICML.cc/2026/Conference — ICML 2026 regular_

### Official Review · Reviewer_M5X4 · 2026-03-09

**Soundness:** 3
**Presentation:** 3
**Significance:** 3
**Originality:** 2
**Overall Recommendation:** 3
**Confidence:** 4

**Summary:**

This paper proposes an out-of-distribution (OOD) detection method for diffusion models. The main idea is that models are more likely to learn relevant symmetries on in-distribution (ID) data than away from the data distribution. Therefore, when the model’s behavior is not symmetry-consistent on a test point, that inconsistency can serve as evidence that the point may be OOD.

In this paper, “learning symmetries” means that the diffusion score function behaves consistently under simple transformations of the input. Roughly, if the input is flipped, rotated, or shifted, the score should transform in the corresponding way. The method measures how well this consistency holds and uses failures of this consistency as an OOD signal.

The paper also provides a theoretical argument for why this approach can separate ID from OOD under suitable assumptions. Empirically, it evaluates the method on standard benchmarks and also studies a setting where the evaluation data occupies only a subset of a broader high-density structure.

**Compliance With Llm Reviewing Policy:**

Affirmed.

**Final Justification:**

score is kept, see rebuttal comment.

**Key Questions For Authors:**

•	Was it empirically observed that models break symmetries in off-sample regions? Are there theoretical results that prove this?
	•	Could the paper highlight more clearly what is novel relative to the prior work that also used symmetry breaking for OOD detection (iDECODe)?

**Limitations:**

yes

**Strengths And Weaknesses:**

Pros

1. The main idea is clean and interesting. Using symmetry breaking of the diffusion score as an OOD signal gives a new perspective within diffusion-based OOD detection.
2. The paper provides a theoretical argument connecting symmetry consistency on ID data and symmetry inconsistency off-distribution to successful detection. This makes the method more principled.
3. The discussion (including proofs and experiments) of the setting where the ID data comes from only a smaller subset of a broader high-density structure is valuable. It explains why such points can still be labeled as ID.
4. The empirical results are solid. The method improves over several prior baselines on the tested benchmarks and is competitive overall.
5. The method requires only forward passes and does not require Jacobian computation. This is a practical advantage over some prior diffusion-based methods.

Cons
1. The novelty is narrower than it may first appear. Prior work already used symmetry breaking for OOD detection in classifier settings, so the main novelty here is applying this idea to diffusion score functions.
2. In terms of theory: The paper assumes that the model learns the relevant symmetries on ID data but not in off-sample regions, and uses this to prove that the method can distinguish between ID and OOD. This is a useful formalization, however, for the theoretical argument it already assumes symmetry breaking is a signal that can distinguish ID and OOD. It would be much more interesting to obtain a theoretical understanding of why symmetry breaking in off-sample regions should arise naturally.
3. The practical advantage in computational efficiency is not completely clear. Although the method avoids Jacobian computation, it still requires 16 forward passes to achieve its reported performance, so the overall compute benefit relative to prior work is somewhat ambiguous.

---

> ### Author Rebuttal · Authors · 2026-03-30
>
> We thank the reviewer for the careful reading and thoughtful comments. We agree that the novelty should be positioned more precisely, that the mechanism behind off-sample symmetry breaking should be clarified, and that the compute claim should be stated more carefully.
>
> >Was it empirically observed that models break symmetries in off-sample regions?
>
> In our example (SVHN vs CIFAR100) we do observe stronger symmetry breaking in off-sample regions, but we agree that this was not surfaced explicitly enough in the main paper. In fact, Section F.7 of the appendix already contains a representative score-distribution figure comparing (i) score magnitude, (ii) single-step equivariance residual energy, and (iii) the final **GEPC** score. The key pattern is that the score-magnitude baseline weakly separates ID/OOD, the single-step equivariance residual separates better, and the final time-aggregated **GEPC** score separates best. We will move this representative histogram from the appendix into the main paper (or summarize it explicitly there), since it directly addresses whether symmetry breaking is empirically observed off-sample.
>
> >Are there theoretical results that prove this? It would be much more interesting to obtain a theoretical understanding of why symmetry breaking in off-sample regions should arise naturally.
>
> We agree that the most interesting theoretical question is not only when **GEPC** separates, but why symmetry breaking should arise naturally away from the ID region. We do not claim a universal theorem stating that every off-sample region must exhibit equivariance-breaking. Rather, our point is that such breaking is a natural consequence of how score fields are learned.
>
> Denoising score matching constrains $s_\theta(\cdot,t)$ through expectations under the forward marginals $q_t$. Hence, the loss primarily constrains the score field on high-density regions induced by ID data  and learns its symmetry. Outside these regions, the objective provides much weaker supervision.
> Since the main anomaly detection hypothesis is that ID and OOD distribution should differ, the learned score field does not exhibit symmetry with high probability for OOD samples that are not in ID support.
>
> In addition, Proposition 4.3 provides a complementary cross-backbone mechanism: when the score field is accurate and approximately equivariant near the source manifold, the expected **GEPC** residual increases with distance from that manifold, explaining why equivariance consistency should deteriorate in off-manifold / off-sample regions.
>
> >Could the paper highlight more clearly what is novel relative to the prior work that also used symmetry breaking for OOD detection (iDECODe)?
>
> We will make the distinction from iDECODe much sharper. iDECODe detects equivariance violations of a discriminative / auxiliary-task model through a conformal non-conformity score and emphasizes false detection rate control. GEPC is different in three ways:
> (1) it probes the multi-timestep consistency of a pretrained diffusion score field itself, rather than an auxiliary transformation-prediction or discriminative model;
> (2) it uses ID-only calibration rather than conformal prediction to convert scores into thresholds with coverage guarantees and;
> (3) it is supported by diffusion-specific population analysis (ideal residual, ID upper bounds, OOD lower bounds, distance-to-manifold discussion) and naturally yields spatial equivariance-breaking maps from the score field.
>
> >The practical advantage in computational efficiency is not completely clear.
>
> Finally, we agree that the practical compute claim should be phrased more carefully. Our point is not that **GEPC** is uniformly cheaper than all prior methods, but that it reaches a favorable forward-only operating point without resorting to the Jacobian. We will state this more explicitly and add the lighter **GEPC-lite** variant, instantiated as the flips-only subgroup under an **$8F{+}0J$** compute budget, which remains competitive (with average AUROC of $0.910$) while reducing compute. In addition, our group ablations show that reduced transform sets can materially affect both performance and held-out ID stability, making the compute / stability trade-off more explicit.
>
> We hope these revisions make the paper’s novelty, empirical evidence, theoretical scope, and computational positioning substantially clearer.

---

> > ### Author Rebuttal · Reviewer_M5X4 · 2026-04-04
> >
> > I thank the authors for their response.
> > I keep the score as is because:
> > - the idea for using symmetry for OOD has already been shown in prior work.
> > - The paper does manage to use symmetries for OOD detection in diffusion models and its adaptation is non-trivial. While the experiments are very good, it is still not convincing that it overcomes prior methods. This would be okay if the idea was completely novel, but since the idea appeared on prior work I don't feel like it's sufficient.
> > - The proof is still assuming symmetry breaking.
> >
> > Note: If one of these three limitations was fully addressed, then the paper might have been a good fit. However, given the current state, I do not see the edge of the paper that will push it in the conference.

---

> > > ### Author Response · Authors · 2026-04-05
> > >
> > > Thank you for this clarification. We understand that the remaining issue is the precise edge of **GEPC** relative to prior symmetry-based OOD work and prior diffusion-OOD methods.
> > >
> > > We would like to make that edge as explicit as possible. **GEPC** does not introduce yet another scalar observable of a pretrained diffusion model. Rather, it probes a structural property of the pretrained multiscale score field itself: transport-equivariance across transformations and noise levels. For a group-invariant forward marginal, the ideal score satisfies
> > >
> > > $$
> > > s_t(P_gx)=P_g s_t(x).
> > > $$
> > >
> > > **GEPC** converts violations of this relation across timesteps into an OOD statistic on a frozen diffusion backbone. In this sense, the diffusion-specific contribution is not "symmetry for OOD" in general, but using multiscale score transport-equivariance itself as the object of the probe.
> > >
> > > On the theory side, we agree that our main separation result is conditional. Our point is narrower: we do not claim a universal theorem that every OOD point must break symmetry. Such a universal statement would in fact be too strong in general, since an OOD distribution may share the same symmetries as the ID distribution and therefore need not induce large equivariance defects. Rather, we explain why degradation of transport-equivariance away from the high-density ID region is naturally permitted by denoising score matching (DSM), rather than merely postulated.
> > >
> > > For each $t$, let the population score be
> > >
> > > $$
> > > s_t^\star(x)=\nabla \log q_t(x),
> > > $$
> > >
> > > and consider the DSM objective
> > >
> > > $$
> > > L_t(s)=E_{x\sim q_t}[||s(x)-s_t^\star(x)||_2^2].
> > > $$
> > >
> > > Now let $h$ be any perturbation of the score field, and recall the equivariance residual
> > >
> > > $$
> > > \Delta_g s(x):=P_g^{-1}s(P_gx)-s(x).
> > > $$
> > >
> > > Then we have the exact identity
> > >
> > > $$
> > > L_t(s_t^\star+h)-L_t(s_t^\star) = E_{x\sim q_t}[||h(x)||_2^2].
> > > $$
> > >
> > > Therefore, if $h$ is concentrated in a region where $q_t$ has small mass, the DSM cost can be arbitrarily small even when $h$ induces a large local change in the equivariance residual. This does not prove that every OOD region must exhibit symmetry breaking; rather, it shows that DSM does not strongly enforce transport-equivariance in low-density regions of $q_t$.
> > >
> > > Our point is therefore not that symmetry breaking is assumed as a black-box discriminator, but that DSM enforces the score where the forward ID marginals place mass and does not determine how transport-equivariance must continue away from the high-density ID region. In this sense, symmetry degradation is a natural consequence of localized supervision. Proposition 4.3 provides the complementary geometric statement: once the residual is small near the source manifold, it grows with distance from that manifold in the cross-backbone regime.
> > >
> > > To make the edge even more concrete, the Gaussian analysis in Sec. 4.1 and the empirical illustration in Appendix Fig. 4 provide an explicit witness that the transport-equivariance residual is not reducible to a standard score-only observable. In particular, there exist Gaussian/linear regimes in which score-based scalar diagnostics are non-separating, while the transport-equivariance residual separates. We do not present this as a universal theorem for all OOD settings, but as a constructive witness that **GEPC** probes information in the same pretrained score field that standard observables do not capture.
> > >
> > > To state the positioning as precisely as possible: **GEPC** does not claim symmetry-based OOD detection in full generality, nor a universal theorem that all OOD samples must break symmetry. Its contribution is to identify a diffusion-specific observable, namely multiscale score transport-equivariance on a pretrained backbone, and to connect its degradation away from the high-density ID region to a principled DSM-based explanation. We believe this clarifies the main edge of the paper relative to prior symmetry-based OOD work and prior diffusion-OOD methods.
> > >
> > > We hope this clarification more directly addresses the reviewer’s concern.

---

### Official Review · Reviewer_rFn4 · 2026-03-11

**Soundness:** 3
**Presentation:** 2
**Significance:** 3
**Originality:** 3
**Overall Recommendation:** 5
**Confidence:** 3

**Summary:**

This paper presents Group-Equivariant Posterior Consistency (GEPC), a training-free method for OOD detection in diffusion models. The presented method measures how the learned score field equivariance holds (equivariance consistency on InD data) or breaks (OoD data) for a finite symmetry group. Concretely, GEPC transforms a noisy input by a group element, evaluates the score on the transformed input, transports the result back, and measures the (squared) residual. GEPC aggregates the squared residuals across group elements and selected time steps, yielding a scalar score for OOD detection. The method requires only forward score evaluations and produces both a scalar confidence score for OOD detection and spatial equivariance-breaking maps. At the theoretical level, the authors provide bounds on the expected residuals and perform experiments on standard 32x32 (ID/OOD) datasets using a single diffusion backbone trained on the CelebA dataset. Additionally, the authors show the scalability of their method in a single diffusion backbone for higher-resolution 256x256 SAR detection.

**Compliance With Llm Reviewing Policy:**

Affirmed.

**Final Justification:**

The rebuttal has resolved my questions and concerns. My score is kept.

**Key Questions For Authors:**

1. FPR95 and AUPR are reported only for the SAR detection in Table 9. Can the authors also report these metrics for the other smaller-scale experiments (32x32 CelebA backbone)?
2. How sensitive is GEPC to the choice of group G? The paper uses a fixed group for all experiments. An ablation removing individual transform types (e.g., using only flips or only rotations) would help assess robustness. Could the authors discuss how practitioners should select G for a new domain/dataset?
3. Could the authors report AUROC variance over at least 5 random seeds (varying the noise $\varepsilon$ in the forward process)?
4. What would happen with the equivariance signal in non-convolutional architectures?

**Limitations:**

The authors discuss computational cost, symmetry assumptions, and reliance on the diffusion backbone. These limitations are honest and consistent with the identified weaknesses.

**Strengths And Weaknesses:**

---

### Strengths

**S1. Novel perspective (Originality).** The idea of using equivariance consistency of a pretrained diffusion score field as an OOD signal appears to be original within the diffusion-based OOD literature.

**S2. Theoretical framework (Soundness).**  The expected residual bounds (Propositions 4.2–4.3) provide a principled separation guarantee rather than purely empirical claims.
The Gaussian mean-shift example (eq. 12) showcases the benefit of the equivariance residual over the score magnitude.

**S3. Computationally lightweight (Significance).**  GEPC requires only forward passes and no Jacobian computation or backbone finetuning. The method allows parallelization over group elements and timesteps.

**S4. Ablations (Soundness).** The paper provides ablations over group elements, time step selection, calibration modes, feature variants, and runtime.


---

### Weaknesses

**W1. Limited scale and diversity of evaluation (Significance/Soundness)**:
- No experiments with higher resolution in standard ODD benchmarks (ImageNet scale as in DiffPath).
- The experiments consider a single backbone trained on a single dataset (CIFAR & LSUN). This is particularly relevant to assess the validity of the assumption (L241, right column) regarding the cross-backbone Proposition 4.3.
- No experiments or discussion on equivariance signal for OOD detection in alternative architectures (beyond Convolutional, such as DiT)

**W2. Equivariance Maps Experimental Design & Results (Presentation & Soundness).** The SAR evaluation reads more like a compelling proof-of-concept demonstration rather than a rigorous experimental contribution to support the authors' claims:
- The evaluation uses a single backbone (trained LSUN-256), challenging the assumption for Proposition 4.3, as in W1.
- The paper presents four qualitative figures (Figures 2 and 5) with a global normalization scheme, and a brief description of the normalization procedure in Appendix G.  A rigorous spatial interpretability evaluation would need: quantitative localization metrics (e.g., IoU), systematic comparison of map quality across timesteps and group elements, analysis of failure cases (when does the map highlight a wrong region?), comparison against other interpretability methods, etc.
- This larger-scale evaluation seems disconnected from the experiments/evaluations and observations in the small-scale regime (CelebA backbone, 32x32 experiments).

**W3. No variance reporting (Soundness).** The proposed method has sources of stochasticity, e.g., via MC noise sampling. Although the authors mention this source of variability (L290-291, left column), the reported performance results do not reflect it.

**W4. Rationale and Guidance for choosing group G (Soundness/Significance).** The rationale for the default set of transforms (group G) is not explained. The authors provide an isolated ablation, but they do not discuss group selection for new domains/datasets or the selection of potential subgroups of transforms and their corresponding performance.

---

> ### Author Rebuttal · Authors · 2026-03-30
>
> We thank the reviewer for the positive assessment. We address the main concerns by adding four elements.
>
> > FPR95 and AUPR are reported only for the SAR detection in Table 9. Can the authors also report these metrics for the other smaller-scale experiments (32x32 CelebA backbone)?
>
> In the revision, we add a appendix table reporting average AUROC, AUPR, and FPR95 over the same 9 standard $32\times32$ ID/OOD pairs under the frozen CelebA-32 shared-backbone protocol:
>
> | Method    | AUROC | AUPR  | FPR95 |
> |-----------|------:|------:|------:|
> | GEPC-lite | 0.910 | 0.907 | 0.265 |
> | GEPC      | 0.908 | 0.904 | 0.274 |
>
> Here, **GEPC-lite** denotes the flips-only subgroup variant under an $8F{+}0J$ compute budget. The full appendix table reports the same metrics for the other shared-backbone baselines (NLL, IC, MSMA, DDPM-OOD, and LMD) as well.
>
> > How sensitive is **GEPC** to the choice of group $G$? The paper uses a fixed group for all experiments. An ablation removing individual transform types (e.g., using only flips or only rotations) would help assess robustness. Could the authors discuss how practitioners should select $G$ for a new domain/dataset?
>
> We agree that the choice of $G$ is important. We therefore add a group-ablation summary with average (across the 3 ood datasets)  AUROC (**A**), AUPR (**P**), FPR95 (**F**), and held-out ID coefficient of variation (ID-**CV**; definition in Section 5.2):
>
> | Group       | SVHN A / P / F / CV         | CIFAR10 A / P / F / CV      |
> |-------------|-----------------------------|-----------------------------|
> | Full        | 0.925 / 0.933 / 0.351 / 0.239 | 0.798 / 0.775 / 0.466 / 0.222 |
> | Flips only  | 0.945 / 0.954 / 0.300 / 0.143 | 0.786 / 0.766 / 0.493 / 0.285 |
> | Shifts only | 0.869 / 0.875 / 0.495 / 1.124 | 0.754 / 0.721 / 0.561 / 0.316 |
> | No shifts   | 0.937 / 0.949 / 0.331 / 0.160 | 0.802 / 0.785 / 0.475 / 0.197 |
>
> The pattern is consistent: shift-only gives the weakest performance and the largest ID-CV, whereas simpler groups such as flips-only or no-shifts (flips + rotations) remain competitive and much more stable. This supports a practical ID-only rule: for a new domain, choose $G$ from a small set of plausible candidate groups using held-out ID data only, prefer groups with low ID-CV, and exclude transforms that clearly violate domain geometry or semantics. The full appendix additionally reports rotations-only, no-flips, and no-rotations, with the same qualitative conclusion.
>
> > Could the authors report AUROC variance over at least 5 random seeds (varying the noise $\epsilon$ in the forward process)?
>
> We also add a 5-seed sensitivity experiment in which the checkpoint, splits, group, timestep selection, and all hyperparameters are fixed, and only the Gaussian forward-noise seed is varied:
>
> | ID$\rightarrow$OOD | AUROC std | AUPR std | FPR95 std |
> |--------------------|----------:|---------:|----------:|
> | SVHN$\rightarrow$C10    | 0.0009 | 0.0005 | 0.0073 |
> | SVHN$\rightarrow$CelebA | 0.0000 | 0.0000 | 0.0000 |
> | SVHN$\rightarrow$C100   | 0.0010 | 0.0011 | 0.0132 |
>
> These deviations are very small, indicating that **GEPC** is stable with respect to Monte Carlo forward noise under the reported configuration.
>
> > No experiments with higher resolution in standard OOD benchmarks (ImageNet scale as in DiffPath).
>
> To partially address scalability, we add a compact large-scale stress test on a frozen ImageNet-64 diffusion backbone:
>
> | Method | SSB-hard | Textures | Mean |
> |--------|---------:|---------:|-----:|
> | MSMA   | 0.591    | 0.718    | 0.655 |
> | GEPC   | 0.640    | 0.718    | 0.679 |
>
> We present this as a diffusion-side scalability check rather than an apples-to-apples comparison against supervised classifier-based OpenOOD post-hoc baselines, but it shows that the equivariance-consistency signal remains informative beyond the $32\times32$ CelebA setting.
>
> > The SAR evaluation reads more like a compelling proof-of-concept demonstration rather than a rigorous experimental contribution to support the authors' claims.
>
> Regarding the SAR section, we agree that the original presentation was closer to a proof-of-concept than to a full interpretability benchmark. We therefore add a classical Reed-Xiaoli (RX) baseline anomaly detector (Reference: Reed & Yu, 1990) on the same HRSID/SSDD splits: HRSID-ship/wake $= 0.573 / 0.552 / 0.613$ and SSDD-ship $= 1.000 / 1.000 / 0.000$ (AUROC / AUPR / FPR95).
>
> > What would happen with the equivariance signal in non-convolutional architectures?
>
> On non-convolutional architectures such as DiT, **GEPC** is not intrinsically tied to CNN backbones. It only requires access to a learned score-related field and a family of input transforms. Thus, the construction still makes sense for DiT-like models as long as one can recover such a field from the model output.
>
> We hope these additions address the reviewer’s concerns by broadening the evaluation, quantifying variance, and providing a concrete empirical guideline for selecting $G$.

---

> > ### Author Rebuttal · Reviewer_rFn4 · 2026-04-01
> >
> > The authors have addressed my concerns. I will increase my score from 4 to 5.

---

> > > ### Author Response · Authors · 2026-04-01
> > >
> > > We are thankful for the positive feedback and for reconsidering the score. We especially thank the reviewer for highlighting the importance of group selection.

---

### Official Review · Reviewer_bc6Q · 2026-03-12

**Soundness:** 2
**Presentation:** 4
**Significance:** 3
**Originality:** 3
**Overall Recommendation:** 5
**Confidence:** 4

**Summary:**

This paper proposes GEPC, a training-free OOD score for diffusion models based on equivariance breaking of the learned score field under a finite transformation group. The idea is interesting and technically clean: instead of using score magnitude or local curvature, the method probes whether the score transforms consistently under flips/rotations/shifts. The paper also provides a population-level analysis and interpretable residual maps.

**Compliance With Llm Reviewing Policy:**

Affirmed.

**Final Justification:**

Although the empirical results are limited, the idea of the paper is novel and interesting to the OOD community. The paper is well written with good presentation. I therefore increase my score to 5 and lean toward acceptance.

**Key Questions For Authors:**

- Could you provide more justification/boundary on group misspecification if the chosen group does not match the data geometry.

- The current experiments are conducted on relatively small-scale datasets, so it remains unclear how diffusion-based OOD methods would compare against strong post-hoc baselines in a more standard large-scale setting such as OpenOOD [1]. I understand that training and evaluating diffusion models on ImageNet-scale data would be substantially more expensive. Still, it would be interesting to discuss whether this limitation could be alleviated by training diffusion models in representation space, as in [2], and then applying GEPC on learned image features rather than raw pixels. Such a setting could make the method more scalable and would help clarify its practical competitiveness against post-hoc approaches.

I would increase my score if my concerns are addressed adequately.

[1] OpenOOD: Benchmarking Generalized Out-of-Distribution Detection

[2] Revisiting Likelihood-Based Out-of-Distribution Detection by Modeling Representations

**Limitations:**

yes

**Strengths And Weaknesses:**

$\textbf{Strengths:}$

- The core idea is novel and conceptually well motivated with Group-Equivariant.

- Great written and presentation. Figure 1 explains the method clearly, and the distinction from curvature- and trajectory-based diffusion OOD scores is well presented.

- The method is appealing in that it is training-free, does not require Jacobians, and yields spatial maps in addition to scalar scores. Reasonable ablation depth.

$\textbf{Weaknesses:}$

- My main concern is that the method’s advantage is not uniformly strong in the main benchmark. In Table 1, GEPC is competitive but not clearly best on average, for example, MSMA and DiffPath achieve higher average AUROC than GEPC. Therefore, the claim of strong empirical superiority should be stated more carefully.

- The method depends on a hand-chosen symmetry group. The paper acknowledges this in the discussion, but this is not a minor detail: if the chosen group does not match the data geometry, GEPC may fail for structural reasons unrelated to OODness.

---

> ### Author Rebuttal · Authors · 2026-03-30
>
> We thank the reviewer for the positive assessment and constructive comments.
>
> > Could you provide more justification/boundary on group misspecification if the chosen group does not match the data geometry.
>
> **GEPC** measures the discrepancy between the score function $s(\mathbf{x})$ and its transformed counterpart $\mathcal{P}_g s(\mathcal{P}_g \mathbf{x})$. When ID data are not invariant to a given transformation $g$, this discrepancy is non-zero. In that case, the **GEPC** score for ID samples can be written as
> $$
> f_I(\mathcal{P}_g) = ||s(\mathbf{x}_I) - \mathcal{P}_g s(\mathcal{P}_g \mathbf{x}_I)||^2 > 0.
> $$
>
> Since a kernel density estimator (KDE) is trained to model the distribution of **GEPC** scores on ID data, the corresponding OOD scores
> $$
> f_O(\mathcal{P}_g) = ||s(\mathbf{x}_O) - \mathcal{P}_g s(\mathcal{P}_g \mathbf{x}_O)||^2
> $$
> are expected to deviate from this learned distribution.
> This behavior can be illustrated with the effect of a $180^\circ$ rotation by considering the group $G = \{ \mathrm{Id}, -\mathrm{Id} \}$. Let
> $$
> \mathbf{x}_I \sim \mathcal{N}(0, I), \quad
> \mathbf{x}_O \sim \mathcal{N}(\mu, I).
> $$
> In this setting, the ID distribution is invariant to the transformation $-\mathrm{Id}$. Using the score of a Gaussian distribution, one can show (see Sections 4 and C.1) that the **GEPC** score is equal to $0$ for ID samples, and $2||\mu||^2$ for OOD samples, enabling discrimination between the two distributions.
>
> Now consider a misspecified setting where the ID distribution is not invariant to the chosen transformation, e.g.,
> $$
> \mathbf{x}_I \sim \mathcal{N}(\epsilon, I).
> $$
> In this case, the **GEPC** score for ID samples becomes $2||\epsilon||^2$, while the OOD score remains $2||\mu||^2$. As long as $\epsilon \neq \mu$, the two distributions remain distinguishable. However, when $||\epsilon||$ approaches $||\mu||$, the separation becomes less pronounced compared to the invariant case.
>
> Overall, this example shows that choosing a group that does not perfectly match the data geometry may degrade the separation between ID and OOD data, but **GEPC** remains a valid approach for OOD detection as long as the score induces different responses for ID and OOD data.
>
>
> > The method depends on a hand-chosen symmetry group. [...] if the chosen group does not match the data geometry, **GEPC** may fail
>
>
> To address this, we add a group-ablation summary with average AUROC (across the 3 ood datasets) and held-out ID coefficient of variation (ID-CV, Section 5.2):
>
> | Group       | SVHN (AUROC) | SVHN (ID-CV) | CIFAR10 (AUROC) | CIFAR10 (ID-CV) |
> |-------------|-------------:|-------------:|----------------:|----------------:|
> | Full group  | 0.925        | 0.239        | 0.798           | 0.222           |
> | Flips only  | 0.945        | 0.143        | 0.786           | 0.285           |
> | Shifts only | 0.869        | 1.124        | 0.754           | 0.316           |
> | No shifts   | 0.937        | 0.160        | 0.802           | 0.197           |
>
> The pattern is consistent: shift-only yields the highest ID-CV and weakest OOD performance, while simpler groups (flips-only, no-shifts) remain competitive and more stable. This supports an ID-only rule: select $G$ from a small candidate set using held-out ID data, prefer low ID-CV, and exclude transforms violating domain geometry or semantics.
>
> > My main concern is that the method’s advantage is not uniformly strong in the main benchmark.
>
> We agree and moderate the claim. **GEPC** does not uniformly dominate prior diffusion-OOD methods; rather, it offers a distinct operating point: training-free, forward-only (no JVPs), interpretable equivariance-breaking maps, and competitive under a strict shared-backbone protocol. We also add **GEPC-lite** (flips-only version of **GEPC**, $8F{+}0J$, AUROC $0.910$) to clarify the compute/performance trade-off.
>
> > The current experiments are conducted on relatively small-scale datasets
>
> To address scalability, we add a compact large-scale test on a frozen ImageNet-64 diffusion backbone with ImageNet-style OOD data and a matched comparison with MSMA:
>
> | Method | SSB-hard | Textures | Mean |
> |--------|---------:|---------:|-----:|
> | MSMA   | 0.591    | 0.718    | 0.655 |
> | GEPC   | 0.640    | 0.718    | 0.679 |
>
> We present this as a diffusion-side scalability check rather than a direct comparison to supervised OpenOOD baselines, showing the equivariance-consistency signal remains informative beyond $32\times32$ CelebA.
>
> > this limitation could be alleviated by training diffusion models in representation space...
>
> Finally, regarding the reviewer’s suggestion about representation-space modeling, we agree that this is a promising direction. In our view, **GEPC** is not intrinsically tied to raw pixels: the same equivariance-consistency principle could in principle be applied to diffusion models trained in a learned representation space, which may be a route for future work.
>
> We hope these revisions clarify the reviewer concerns.

---

> > ### Author Rebuttal · Reviewer_bc6Q · 2026-03-31
> >
> > My concerns have been addressed by the authors. Given the quality of the paper, although the empirical results are limited, the idea itself is interesting to the OOD community. I therefore increase my score to 5 and lean toward acceptance. However, I still suggest that the authors clearly clarify the limitations in terms of evaluation significance and discuss potential future extensions.

---

> > > ### Author Response · Authors · 2026-04-01
> > >
> > > We thank the reviewer for the positive feedback and for reconsidering the score. We will clarify the limitations of our empirical evaluation and more clearly discuss potential future extensions of the method.

---

### Official Review · Reviewer_pnWZ · 2026-03-12

**Soundness:** 2
**Presentation:** 2
**Significance:** 2
**Originality:** 3
**Overall Recommendation:** 4
**Confidence:** 4

**Summary:**

This paper proposes GEPC, a diffusion-based method for out-of-distribution detection that leverages invariance under transformations. Specifically, given a test image, for each timestep and transformation from a selected set, GEPC compares the score predicted from the test image at that timestep, and the invert-transformed score of the transformed test image at that timestep. It computes the residual norms between the two and aggregates across transformations and timesteps to obtain a final score. GEPC then estimates the distribution of this score from in-domain data, and measures the typicality of a test image’s score for OOD detection. Evaluations are performed across both standard datasets (CIFAR, SVHN, CelebA) with a diverse selection of existing baselines, and a higher resolution setting involving SAR imagery.

**Compliance With Llm Reviewing Policy:**

Affirmed.

**Final Justification:**

The rebuttal has resolved my questions and I have raised my score to 4.

**Key Questions For Authors:**

* Would be helpful to clarify the diffusion-based baselines (see Soundness Weaknesses), specifically:
    * Are the numbers for NLL, IC - diffusion, MSMA, DDPM-OOD and LMD in Table 1 all computed with the CelebA diffusion model, regardless of the ID dataset?
    * If so, how are the OOD scoring done for these methods? Because these methods operate under the assumption that the generative model is trained on in-domain, thus having higher likelihood / can better reconstruct ID data. Applying CelebA model to CIFAR or SVHN as in-domain dataset would break this assumption.

* Would be helpful to see baseline comparison for the SAR experiments.

* Would be helpful to have more clear and in-depth discussion on the strengths and applicability of the proposed method, particularly addressing the empirical performance aspect.

**Limitations:**

yes

**Strengths And Weaknesses:**

[Soundness]

Strengths:
* The paper provides extensive theoretical justifications.
* The experimental setups are appropriate for the task, covering established and diverse datasets, and relevant baseline methods.

Weaknesses:

* Clarifications on the baselines: For these baselines: NLL, IC - diffusion, MSMA, DDPM-OOD and LMD, are all results in Table 1 computed from the same CelebA diffusion model? L338-339 mentions that training-free diffusion-based scores are computed from the same CelebA backbone, and Table 1 puts them under this category (although in L358-369 only DiffPath and SCOPED are mentioned).
    * NLL, IC - diffusion, MSMA, DDPM-OOD and LMD do rely on the assumption that the diffusion model itself was learned from ID data, at least in their original settings. If the same CelebA model is used even for CIFAR10 or SVHN as ID, directly applying their original OOD score calculation does not make sense.
    * For IC (diffusion), it’s observed that for all the inverse pairs, the AUROC roughly sums to 1 (e.g. CIFAR10 vs. SVHN - 0.921 and SVHN vs. CIFAR10 - 0.080; CIFAR10 vs. CelebA 0.516 and CelebA vs. CIFAR10 0.485; SVHN vs. CelebA 0.028 and CelebA vs. SVHN 0.972). This would fit the pattern if the same CelebA model is used, because IC’s OOD score for a fixed image and diffusion model would be the same and reversing the pairs would just be flipping the labels, leading to 1-AUROC. However, this is not consistently observed for other baselines (NLL, MSMA, DDPM-OOD and LMD).
   * It would be helpful to just double check the baseline implementations generally, and be more detailed about the implementations and any adaptations or changes made.
* For the Radar SAR OOD detection experiment, there is no comparison with any baselines. Although the performance is good, without comparison with baselines, it’s hard to tell whether the setting itself is saturated or the method is effective.

[Presentation]

Overall the presentation is fine. Some parts of the supplementary are a bit rough, e.g. Table overflowing (Supp. Table 2 and 5), and too many significant digits in Supp. Table 3-8.

[Significance / Originality]

Strength:
* The idea of exploring group invariance as a signal for OOD detection is interesting and novel.

Weakness:

* The proposed method is actually fairly complicated with many components and hyperparameters. However, empirically, the proposed method does not provide a noticeable performance gain over existing methods (e.g. underperforming DiffPath in both performance and compute). It is not super clear where the benefit or necessity of the proposed method lies.

---

> ### Author Rebuttal · Authors · 2026-03-30
>
> We thank the reviewer for identifying an ambiguity in our original presentation. We agree that our wording around Table 1 was too strong.
>
> > Clarification of diffusion baselines and backbone protocol
>
> In the revision, we reorganize Table 1 into three blocks: (1) prior non-diffusion baselines (trained per ID; from prior work: IC, IGEBM, VAEBM, Improved CD, DoS, WAIC, TT, LR), (2) prior diffusion baselines (from prior work; generally ID-specific: NLL, IC, MSMA, DDPM-OOD, LMD), and (3) shared-backbone diffusion methods (single frozen CelebA-32 backbone: DiffPath, SCOPED, **GEPC-lite**, **GEPC**. **GEPC-lite** is the flips-only version of **GEPC** with an average AUROC of 0.910 and a compute budget of 8F). Thus, the reviewer is correct: NLL / IC / MSMA / DDPM-OOD / LMD should not all be read as belonging to one uniform shared-backbone block.
>
> To address this directly, we add a second main-table comparison where all methods use the same frozen CelebA-32 checkpoint and differ only by their test-time statistic or reconstruction rule:
>
> | Method    | C10(ID)            | SVHN(ID)           | CelebA(ID)         | Avg   | F+J    |
> |-----------|--------------------|--------------------|--------------------|-------|--------|
> | NLL       | 0.066/0.180/0.506  | 0.933/0.859/0.924  | 0.822/0.141/0.798  | 0.581 | 4000F  |
> | IC        | 0.703/0.072/0.549  | 0.296/0.023/0.343  | 0.929/0.977/0.935  | 0.536 | 4000F  |
> | MSMA      | 0.767/0.818/0.534  | 0.884/0.960/0.892  | 0.952/0.991/0.948  | 0.861 | 10F    |
> | DDPM-OOD  | 0.076/0.114/0.473  | 0.936/0.697/0.915  | 0.895/0.310/0.861  | 0.586 | 364F   |
> | LMD       | 0.366/0.713/0.572  | 0.634/0.778/0.681  | 0.284/0.223/0.385  | 0.515 | $10^4$F |
> | DiffPath  | 0.910/0.897/0.590  | 0.939/0.979/0.953  | 0.998/1.00/0.998   | 0.918 | 10F    |
> | SCOPED    | 0.814/0.940/0.477  | 0.971/0.996/0.959  | 0.925/0.994/0.962  | 0.892 | 2F+2J  |
> | **GEPC-lite** | 0.821/0.990/0.548  | 0.906/1.00/0.928   | 1.00/1.00/0.999    | 0.910 | 8F     |
> | **GEPC**      | 0.842/0.999/0.554  | 0.880/1.00/0.897   | 1.00/1.00/1.00     | 0.908 | 16F    |
>
> We now clarify the likelihood baselines more explicitly. NLL uses the default `improved-diffusion` likelihood routine `total_bpd`. IC is implemented as $IC(x)=NLL_{bpd}(x)-C_{bpd}(x)$, where $C_{bpd}(x)$ is a lossless compression term normalized in bits per dimension. There is no KDE/GMM fitting or ID-specific calibration in this implementation.
>
> > For IC (diffusion), it’s observed that for all the inverse pairs, the AUROC roughly sums to 1
>
> The reviewer's observation is correct. In the new table, under a single frozen backbone, IC is a fixed-image statistic; thus, reversing a pair approximately flips the labels and yields $\mathrm{AUROC} \approx 1 - \mathrm{AUROC}$. The same observation applies to NLL, DDPM-OOD, and LMD, as they assume the generative model is trained on in-domain data.
>
> > and be more detailed about the implementations and any adaptations or changes made
>
> We also add a dedicated appendix subsection describing the precise implementations of NLL, IC, MSMA, DDPM-OOD, and LMD.
>
> > Would be helpful to see baseline comparison for the SAR experiment
>
> We agree. We therefore add the classical Reed-Xiaoli (RX) anomaly detector, fitted on clutter-only ID-train patches and evaluated on the same HRSID / SSDD splits as **GEPC**:
>
> | OOD split       | AUROC | AUPR  | FPR95 |
> |-----------------|------:|------:|------:|
> | HRSID-ship/wake | 0.573 | 0.552 | 0.613 |
> | SSDD-ship       | 1.000 | 1.000 | 0.000 |
>
> (References: Reed & Yu, 1990; Muzeau et al., IGARSS 2023.)
>
> > Would be helpful to have more clear and in-depth discussion on the strengths and applicability of the proposed method, particularly addressing the empirical performance aspect.
>
> We will moderate the empirical claims. Our claim is not uniform superiority over all prior methods. Rather, **GEPC** provides a distinct operating point: it is training-free, forward-only (no JVPs), produces interpretable equivariance-breaking maps, and remains competitive under a strict shared-backbone protocol. We make this clearer in two ways. First, the Gaussian mean-shift example (see Sections 4 and C.1) already shows that the equivariance residual can separate regimes where score magnitude alone cannot. Second, we add **GEPC-lite**, the flips-only subgroup under an **$8F{+}0J$** budget, which remains highly competitive (average AUROC 0.910) and makes the compute/performance trade-off explicit. We also add group ablations showing that reduced transform sets can stay competitive while materially changing held-out ID stability.
>
> > Some parts of the supplementary are a bit rough, e.g. Table overflowing (Supp. Table 2 and 5), and too many significant digits in Supp. Table 3-8
>
> We reorganize the appendix, fix table overflows, and report values with three significant digits for readability.
>
> We hope these revisions resolve the protocol ambiguity, strengthen the SAR evidence, and clarify the practical scope of **GEPC**.

---

> > ### Author Rebuttal · Reviewer_pnWZ · 2026-04-03
> >
> > Thank the authors for the rebuttal. It has resolved my questions and I have raised my score to 4.

---

> > > ### Author Response · Authors · 2026-04-04
> > >
> > > We thank the reviewer for the careful reading, the constructive feedback, and for reconsidering the score. We are glad that the rebuttal helped clarify the main protocol questions and addressed the concerns. We especially thank the reviewer for helping us improve the presentation and clarify the empirical claims.

---

### Decision · Program_Chairs · 2026-04-30

**Decision:**

Accept (regular)

**Comment:**

This paper introduces GEPC, a diffusion model based ood detectors. It measures how consistently the learned score transforms under a finite group. The idea is novel and the theoretical results provide intuitions on how this method works. The authors are recommended to include results on commonly-used ood detection datasets in addition to current ones in the final version to further strengthen this paper.